# EAT v1.0.0: a 1D testbed for physical-biogeochemical data assimilation in natural waters

Jorn Bruggeman[1,*], Karsten Bolding[1], Lars Nerger[2], Anna Teruzzi[3], Simone Spada[3], Jozef Skákala[4,5], Stefano Ciavatta[6]

[1]Bolding & Bruggeman ApS, Asperup, 5466, Denmark
[2]Alfred-Wegener-Institut Helmholtz-Zentrum für Polar- und Meeresforschung, Bremerhaven, 27570, Germany
[3]National Institute of Oceanography and Applied Geophysics - OGS, Trieste, 34010, Italy
[4]Plymouth Marine Laboratory, Plymouth, PL1 3DH, UK
[5]National Centre for Earth Observation, Plymouth, PL1 3DH, UK
[6]Mercator Ocean International, Toulouse, 31400, France

*Correspondence to*: Jorn Bruggeman ([jorn@bolding-bruggeman.com](jorn@bolding-bruggeman.com))

**Abstract.** Data assimilation (DA) in marine and freshwater systems combines numerical models and observations to deliver the best possible characterisation of a water body's physical and biogeochemical state. DA underpins the widely used 3D ocean state reanalyses and forecasts produced operationally by e.g. the Copernicus Marine Service. The use of DA in natural waters is an active field of research, but testing new developments in realistic setting can be challenging, as operational DA systems are demanding in terms of computational resources and technical skill. There is a need for testbeds that are sufficiently realistic but also efficient to run and easy to operate. Here, we present the Ensemble and Assimilation Tool (EAT): a flexible and extensible software package that enables data assimilation of physical and biogeochemical variables in a one-dimensional water column. EAT builds on established open-source components for hydrodynamics (GOTM), biogeochemistry (FABM) and data assimilation (PDAF). It is easy to install and operate, and flexible through support for user-written plugins. EAT is well suited to explore and advance the state-of-the-art in DA in natural waters thanks to its support for (1) strongly and weakly coupled data assimilation, (2) observations describing any prognostic and diagnostic element of the physical-biogeochemical model, and (3) estimation of biogeochemical parameters. Its range of capabilities is demonstrated with three applications: ensemble-based coupled physical-biogeochemical assimilation, the use of variational methods (3D-Var) to assimilate sea surface chlorophyll, and the estimation of biogeochemical parameters.

## 1. Introduction

To understand and predict the ocean's capacity for carbon sequestration, its ability to supply food, and its response to climate change, we need to know the ocean state: its physical and biogeochemical properties from surface to sea floor. Our ability to directly observe this state is limited: satellites have extensive, often global, geographic coverage but only observe the ocean surface, while platforms such as moorings and gliders may operate throughout the water column but are limited to specific sites or regions. Our best knowledge of the ocean state is obtained through data assimilation (DA), which combines numerical models and observations to deliver the best possible characterisation of ocean variables. This approach underpins, for instance, the multidecadal reanalyses and ten-day forecasts produced by the Copernicus Marine Service (CMEMS; [https://marine.copernicus.eu](https://marine.copernicus.eu)).

Data assimilation in marine systems has a long history and is widely used in operational setting (Brasseur et al., 2009), but the field is still under active development (Moore et al., 2019; Carrassi et al., 2018). A first example: the arrival of autonomous observing platforms such as Biogeochemical Argo (Roemmich et al., 2019) and the development of new remote sensing products (Brewin et al., 2021) has increased spatial coverage and added new types of observations, notably for ocean biogeochemistry. Extensive experiments are needed to assess how best to integrate these observations in DA pipelines (Skákala et al., 2021; Ford, 2021; Teruzzi et al., 2021a). For instance, coupled assimilation of physical and biogeochemical variables shows promise but also comes with challenges: it can improve (Goodliff et al., 2019) or degrade assimilation results (Park et al., 2018), depending on model and observation specifics. Second, new and enhanced algorithms for blending model simulations and observations are continuously being developed, with established schemes such as 3D-Var and the Ensemble Kalman filter being supplemented by hybrid variational-ensemble schemes and the Particle Filter (Van Leeuwen et al., 2015). Third, extending data assimilation to estimate (biogeochemical) parameters (in addition to model

state) promises to help understand model deficiencies and to improve parametrisations. However, extensive experiments are needed to evaluate the viability of this approach, along with practical aspects such as the regionalized setting of estimated parameters (Brankart et al., 2012). In short, the many promising developments in ocean data assimilation will require rigorous experiments before they can be integrated in established DA systems based on 3D models.

Unfortunately, 3D data assimilation systems are computationally expensive and time-consuming to run. This applies in particular to ensemble-based methods that require potentially order of one hundred of simultaneous model simulations. As computational resources are limited, it is typically not feasible to perform extensive experiments with operational DA systems. A second obstacle is that these systems are technically complex, requiring considerable high-performance computing and programming skills to develop, modify and operate. The number of people capable of operating them is therefore limited. The consequence of these resource and skill requirements is that there is a considerable lag between the development of new DA theory and methods on the one hand, and their evaluation and application in production systems on the other. There is therefore a clear need for DA testbeds that are both efficient to run and easy to operate.

Here we present the Ensemble and Assimilation Tool (EAT): a flexible and extensible software package that enables data assimilation of physical and biogeochemical variables in a one-dimensional (1D) water column. Water column models are an ideal test beds for data assimilation: they are sufficiently realistic to resolve vertical gradients in temperature, light and biogeochemistry, as well as the role of (turbulent) mixing and its response to meteorological forcing, yet they are efficient enough to allow one year to be simulated in under a minute on a single core of a regular computer workstation. For that reason, experiments in marine data assimilation often focus on 1D, for instance, when testing biogeochemical data assimilation in setups with parametrised physics (Pelc et al., 2012; Eknes and Evensen, 2002; Simon and Bertino, 2012; Bertino et al., 2003), offline physics (Lenartz et al., 2007), and online physics (Torres et al., 2006, 2020; Hoteit et al., 2003; Allen et al., 2003), when assimilating rates as well as state (Mamnun et al., 2022), and when performing state-parameter estimation (Gharamti et al., 2017a, b). EAT is designed specifically to facilitate such experiments. At its core, EAT builds upon established open-source frameworks: water column hydrodynamics are modelled with the General Ocean Turbulence Model (GOTM, Burchard et al., 1999), biogeochemistry with the Framework for Aquatic Biogeochemical Models (FABM, Bruggeman and Bolding, 2014), and data assimilation methods are provided by the Parallel Data Assimilation Framework (PDAF, Nerger and Hiller, 2013). Through these frameworks, users have access to extensive collections of state-of-the-art biogeochemical models and data assimilation schemes. Moreover, these frameworks are also common ingredients of established 3D data assimilation systems; this benefits the transferability of developments from 1D EAT to 3D. In the next sections, we describe the feature set and structure of EAT, along with examples showcasing coupled physical-biogeochemical assimilation, the capability to use both ensemble-based and variational methods, and the ability to perform state-parameter estimation.

## 2. Methods

EAT consists of the Python package "eatpy" which manages the data assimilation filter and the observations to assimilate, and the Fortran executable "eat-gotm" that contains the 1D hydrodynamic-biogeochemical model (Fig. 1). In both components, the core functionality is provided by established, Fortran-based open-source software packages:

- the General Ocean Turbulence Model (GOTM, Burchard et al., 1999) simulates the physics (temperature, salinity, mixing) of the water column.
- the Framework for Aquatic Biogeochemical Models (FABM, Bruggeman and Bolding, 2014) integrates with GOTM to provide a wide range of biogeochemical models.
- the Parallel Data Assimilation Framework (PDAF, Nerger and Hiller, 2013) provides a wide range of data assimilation algorithms.

By wrapping these existing packages, the EAT-specific source code has remained relatively compact ($< 5,000$ lines).

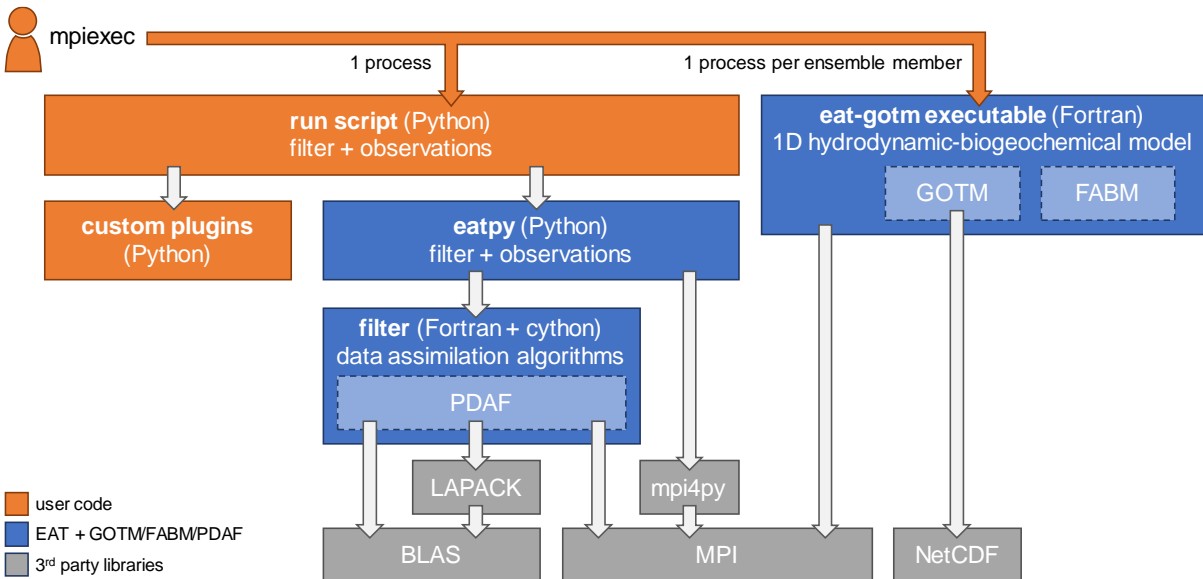

**Figure 1. Components of EAT and its dependencies. Each box indicates a script, compiled executable, Python package, or library; arrows indicate dependencies.**

### 2.1. GOTM: General Ocean Turbulence Model

The General Ocean Turbulence Model (GOTM, Burchard et al., 1999) is a 1D water column model that has been actively developed over the last 25 years. GOTM is written in Fortran 90. It was originally developed to test and compare different turbulence closure models under identical conditions, e.g., forcing and numerics. Accordingly, GOTM comes with a comprehensive library of turbulence closure schemes. In addition, it has over time been extended to become increasingly flexible and configurable (Burchard et al., 2006). As part of this, GOTM has been coupled to FABM (see below) to support coupled simulations with a wide variety of biogeochemical models. GOTM has numerous applications in the ocean and in lakes, several of which includes data assimilation (Torres et al., 2020, 2006; Mattern et al., 2010; Bagniewski et al., 2011; Gharamti et al., 2017b).

GOTM uses a single text file in YAML format (https://yaml.org/), *gotm.yaml*, to describe the physical model configuration. This file in turn can point to additional forcing files, which are text-based and use a format for a time-series of vertical profiles for depth-explicit variables (e.g., temperature, salinity), or a format for time-series of depth-independent variables (e.g., meteorological forcing). All output is written in NetCDF format. The model state, which includes biogeochemistry if active, is written to and optionally read from a restart file; this also uses the NetCDF format.

### 2.2. FABM: Framework for Aquatic Biogeochemical Models

The Framework for Aquatic Biogeochemical Models (FABM, Bruggeman and Bolding, 2014) is a generic Fortran-based framework in which models for marine and freshwater biogeochemistry can be implemented. Several comprehensive models have been implemented in FABM, including those used by most CMEMS Monitoring and Forecasting Centers (MFCs):

- ERSEM (Atlantic European North West shelves) (Butenschön et al., 2016)
- BFM (Mediterranean Sea) (Vichi et al., 2020)
- ECOSMO (Arctic Ocean) (Daewel and Schrum, 2013; Yumruktepe et al., 2022)
- PISCES (global and Iberian Biscay Irish Sea) (Aumont et al., 2015)
- ERGOM (Baltic Sea) (Leibniz Institute for Baltic Sea Research, 2015)

Porting of additional CMEMS biogeochemical models (e.g., BAMHBI for the Black Sea) is currently undertaken in the project NECCTON (https://www.neccton.eu).

FABM integrates with a large variety of hydrodynamic models: couplers have been developed for NEMO, HYCOM, ROMS, GETM, FVCOM, SCHISM, GOTM, among others. FABM-based biogeochemical models analysed in 1D GOTM water columns using EAT are directly available in all these hydrodynamic models, which notably covers all CMEMS MFCs (HYCOM for the Arctic Ocean, NEMO for the other domains).

Accordingly, our understanding of the controllability of BGC models developed by using EAT translates readily to production-ready 3D models.

FABM is configured through a single text file in YAML format, *fabm.yaml*. This file specifies which
biogeochemical processes are active during a simulation, their parameterisation, and default initial values for all state variables. This default initial value is constant across the model domain, but this can be overridden by the hydrodynamic model. For instance, when FABM is coupled to GOTM, depth-explicit initial values or biogeochemical variables can be read from restarts, or specified by pointing to a text file with profiles in *gotm.yaml*.

**2.3.     PDAF: Parallel Data Assimilation Framework**
The parallel data assimilation framework (PDAF, https://pdaf.awi.de, Nerger and Hiller, 2013; Nerger et al., 2005) is a model-agnostic framework for data assimilation. PDAF provides different ensemble filters and smoothers as well as variational methods. In addition, PDAF provides ensemble diagnostics. As a framework it provides support to convert numerical models into models simulating an ensemble of model state realizations.
The data assimilation methods are implemented in a generic way, allowing PDAF to be applied in various modelling applications like ocean physics (Brüning et al., 2021) and sea ice (Mu et al., 2019), ocean biogeochemistry (Goodliff et al., 2019; Pradhan et al., 2019), hydrology (Kurtz et al., 2016), geodynamo (Fournier et al., 2013) and geodynamics (Schachtschneider et al., 2022) or transport dynamics in the atmosphere (Pardini et al., 2020). PDAF is implemented in Fortran 95 with some functions of Fortran 2003 and uses
parallelization with the Message Passing Interface (MPI, Gropp et al., 1994) and OpenMP. It is suited for small applications or toy models, but also high-dimensional models that run on several thousand processor cores (e.g., Kurtz et al., 2016; Nerger et al., 2020). PDAF consist of a core program library and templates for case-specific functions, which build the basis for the implementation for a particular model. The structure of PDAF provides a clear separation of concerns in between the data assimilation method, the model, and observations that are
assimilated.

While PDAF supports both offline and online coupling (Nerger and Hiller, 2013; Nerger et al., 2020), EAT uses online coupling to connect the model to the DA framework: the model state is updated as part of the data assimilation step (analysis) while the simulation remains running. EAT stores the ensemble state internally in an array, which is synchronized with the active GOTM-FABM processes before and after the DA update. PDAF
exposes numerous configuration options, which include the type of data assimilation filter to use as well as various filters-specific settings. EAT enables the user to set these configuration options in a Python run script. Internally, these options are then forwarded to PDAF functions.

**2.4.     Implementation**
The data assimilation core of EAT is the "eatpy" Python package, which includes the filter algorithms as well as
the logic for ensemble generation and observation handling. The user interacts with this package by writing compact Python scripts that generate the model ensemble (Fig 2) and that run the data assimilation experiment (Fig 3). This scripting approach allows the user to retain full control: it provides access to all DA configuration settings, but also makes it straightforward to insert custom code, for instance, to introduce new ensemble generation methods or variable transformations. At run time, a data assimilation experiment combines the user's
Python run script and a number of instances of the coupled hydrodynamic-biogeochemical model (Fig 1). This uses the multiple program-multiple data paradigm, with different components (programs) communicating via the Message Passing Interface (MPI) protocol. The user starts a DA experiment using normal MPI syntax, e.g., `mpiexec -n 1 python <RUNSCRIPT> : -n <NENSEMBLE> eat-gotm <EXTRA_ARGS>`. Here, `<RUNSCRIPT>` is the name of the Python script that defines the data assimilation experiment (Fig. 3),
`<NENSEMBLE>` is the number of ensemble members, and `<EXTRA_ARGS>` are additional arguments to pass to the model, e.g., `--separate_gotm_yaml` to indicate that different ensemble members use different configurations, or `--separate_restart_file` to indicate that different members use different initial states.

```
# Vary the physical and biogeochemical configuration across the N ensemble members.
# We apply log-normally distributed scale factors to:
# * wind speeds (x and y components)
# * background mixing (minimum turbulent kinetic energy)
# * maximum growth rates of the two phytoplankton types
# Each member will have different configuration files for physics (gotm.yaml)
# and biogeochemistry (fabm.yaml). The path to the latter is set in the former.
# Configuration files for each member are written when the "with" clause exits.
rng = np.random.default_rng()
gotm = eatpy.models.gotm.YAMLEnsemble("gotm.yaml", N)
fabm = eatpy.models.gotm.YAMLEnsemble("fabm.yaml", N)
with gotm, fabm:
    gotm["surface/u10/scale_factor"] = rng.lognormal(mean=0.0, sigma=0.2, size=N)
    gotm["surface/v10/scale_factor"] = rng.lognormal(mean=0.0, sigma=0.2, size=N)
    gotm["turbulence/turb_param/k_min"] *= rng.lognormal(mean=0.0, sigma=0.2, size=N)
    gotm["fabm/yaml_file"] = fabm.file_paths
    fabm["instances/phy/parameters/mumax0"] *= rng.lognormal(mean=0.0, sigma=0.2, size=N)
    fabm["instances/dia/parameters/mumax0"] *= rng.lognormal(mean=0.0, sigma=0.2, size=N)

# Vary the initial state across the N ensemble members, using log-normally distributed
# scale factors drawn for each member and variable.
# Restart files for each member are written when the "with" clause exits.
restart = eatpy.models.gotm.RestartEnsemble("restart.nc", N)
with restart:
    for name, values in restart.template.items():
        shape = (N,) + (1,) * values.ndim
        scale_factor = rng.lognormal(mean=0.0, sigma=0.2, size=shape)
        restart[name] = values * scale_factor
```

**Figure 2. Example Python code for generating a data assimilation ensemble with members differing in the parameterisation of the physical model (gotm.yaml), the biogeochemical model (fabm.yaml), and the initial state (restart.nc). Standard numpy functions are used to draw the scale factor of each parameter and state variable from lognormal probability distributions, using an median scale factor of 1 (a mean of 0 in log-transformed space).**


```
import eatpy

# Make the model diagnostic for total chlorophyll available by adding it to the model state.
# This enables us to assimilate chlorophyll observations.
experiment = eatpy.models.GOTM(diagnostics_in_state=["total_chlorophyll_calculator_result"])

# Set up ensemble data assimilation using the Error Subspace Transform Kalman Filter
# (Nerger et al., 2012; https://doi.org/10.1175/MWR-D-11-00102.1)
filter = eatpy.PDAF(eatpy.pdaf.FilterType.ESTKF)

# Identify biogeochemical state variables by checking for an underscore in their name.
# (FABM variable names contain at least one underscore; GOTM physical variable names do not)
bgc_variables = [v for v in experiment.variables if "_" in v]

# Restrict the filter to operating on temperature, salinity and all biogeochemical state variables.
# Notably, other physical variables such as water velocities and turbulent quantities are
# thus not affected by assimilation.
experiment.add_plugin(eatpy.plugins.select.Select(include=["temp", "salt"] + bgc_variables))

# Log-transform all biogeochemical variables. Any associated observations have already been
# log-transformed in preprocessing (therefore, transform_obs=False)
experiment.add_plugin(eatpy.plugins.transform.Log(*bgc_variables, transform_obs=False, minimum=1e-12))

# Link remotely sensed surface temperature and chlorophyll observations to their model
# equivalents. In both cases, the value in the top layer of the model is used.
# This is the *last* model layer in GOTM, specified by index -1.
experiment.add_observations("temp[-1]", "cci_sst.dat")
experiment.add_observations("total_chlorophyll_calculator_result[-1]", "cci_chl.dat")

# Run the experiment
experiment.run(filter)
```

**Figure 3. An example run script, written in Python, for an EAT experiment in which sea surface temperature and chlorophyll are assimilated.**Click or tap here to enter text.

Under the hood, EAT builds on numerous software components. These include the GOTM, FABM and PDAF Fortran codes that are distributed with EAT, as well as established Python packages (e.g., numpy, mpi4py). In turn, these components depend on third-party libraries such as MPI, BLAS, LAPACK, and NetCDF. It can be challenging to assemble and compile these codes and dependencies from scratch. To avoid this becoming a bottleneck, EAT uses the conda package manager (https://conda.io) to set up its compilation and execution environment. Conda pulls in all required packages (runtime dependencies and any necessary compilers) with a single command and further ensures the versions of these components are compatible. EAT offers three conda-based installation options: (1) the user installs a pre-compiled EAT package with a single command that also pulls in all runtime dependencies, (2) the user creates the compilation and execution environment with a single command, and then uses this to compile EAT him/herself. This is appropriate if the user wants to integrate custom Fortran code, in particular, additional FABM-based biogeochemical models, (3) the user sets up a minimal execution environment with a single command, but uses pre-installed compilers to build EAT against pre-installed libraries (MPI, BLAS, LAPACK, NetCDF). This is appropriate for high-performance computing (HPC) systems, which typically have optimized compilers and libraries installed. These three options work on all major platforms – Linux, Mac and Windows. Together, they cover a wide range of use cases: from rapid installation on student laptops for a workshop or course, to custom tailored installations on HPC clusters that will run time-consuming experiments (large ensembles, long simulations, multiple configurations or scenarios, computationally expensive biogeochemistry).

When using ensemble-based or hybrid data assimilation methods, the first step in running an EAT experiment is to generate the ensemble. Ensemble members can differ in their initial state (the "restart"), and in their physical and biogeochemical configuration (*gotm.yaml* and *fabm.yaml*, respectively), which includes parameter values as well as forcing files. EAT includes logic to manipulate GOTM-FABM's NetCDF-based restart files and YAML-based configuration files (Fig 2). For restart files, EAT provides read/write access to all variables that are part of the model state (physical and biogeochemical, depth-explicit and depth-independent). For configuration files, EAT provides read/write access to every setting, independent of its data type (floating-point, integer, Boolean, string). Thus, it is possible to vary real-valued configuration parameters across the ensemble, as well as directing different ensemble members to different forcing files (string-valued paths in the configuration file). Common perturbation strategies, e.g., scaling the original value(s) of a state variable or

parameter with some factor drawn from some user-selected distribution, can be implemented with a single line of code per variable. As EAT provides full access to the spatially explicit initial state and model configuration, more complex strategies can be implemented as well, for instance, ones that impose spatial (vertical) correlations (Evensen, 2003) or cross-correlations among variables and/or parameters.

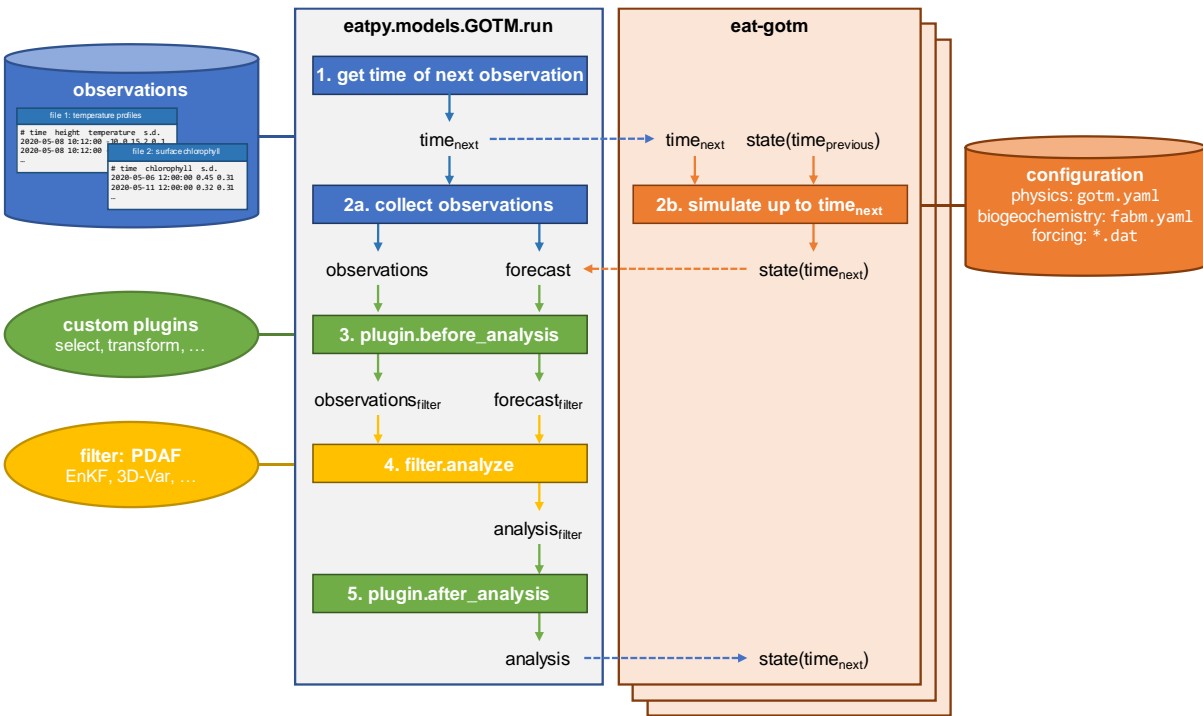

**Figure 4. The data assimilation cycle in EAT, showing the flow of information through the eatpy.model.GOTM.run routine, responsible for observation handling and analysis, and the eat-gotm model component. Dashed arrows indicate inter-process communication between eatpy.model.GOTM.run (1 process) and eat-gotm (1 process per ensemble member), via MPI. After the analysis state is sent back to the model instances (bottom), the cycle repeats with the updated state taking the role of state(time$_{previous}$) on the eat-gotm side.**

The runtime data assimilation cycle in EAT is depicted in Fig 4. The eatpy package exchanges information with one or more instances of the coupled hydrodynamic-biogeochemical model, GOTM-FABM. Assimilation happens *online*: each of the model instances is preserved in between assimilation cycles, with information passed via MPI rather than via restart files. Information that is passed includes the time till which to simulate and the model state before and after analysis. This model state includes all prognostic fields associated with

physics (e.g., temperature, salinity, horizontal velocities, turbulent quantities), pelagic biogeochemistry (e.g., nutrients, plankton), and biogeochemical variables at the surface and bottom interfaces (e.g., variables associated with benthos/sediments). Furthermore, at the direction of the user (in the run script), this state can be augmented with any physical and biogeochemical diagnostic field available within GOTM-FABM (e.g., total chlorophyll, primary production) and any biogeochemical parameter. The latter enables parameter-state

estimation, in which the selected parameter(s) become time-varying and are estimated along with the model state.

A complete data assimilation experiment requires files with observations: one text file per observed variable, with each line describing observation time, observed depth (only for depth-explicit observations), observed value, and its standard deviation in tab-separated value (TSV) format. This format was chosen over structured

formats such as YAML because it enables EAT to read in new observations on-demand while the simulation progresses, instead of having to parse each observation file in its entirety upon start-up. Observations are linked to model variables (state variables and diagnostics) in the user's run script. It is also possible to perform ensemble-only simulations in which no observations are assimilated; such experiments are often used as control in DA studies. They are performed simply by executing the model (eat-gotm) only: `mpiexec -n`

`<NENSEMBLE> eat-gotm <EXTRA_ARGS>`. Finally, it is also possible to run the model stand-alone, without MPI; in this case, it behaves exactly as the original GOTM-FABM model would.

```python
# Plugin that calculates chlorophyll averaged over the first optical depth
# To use it:
# * Adapt variable names below to your biogeochemical model (here: PISCES)
# * Make modelled chlorophyll and light are available to EAT by calling eatpy.models.GOTM
#   with argument diagnostics_in_state=[CHL_NAME, LIGHT_NAME]
# * Link the new chlorophyll metric to observations with
#   experiment.add_observations(CHL_NAME + "_1OD", <FILE>)
CHL_NAME = "total_chlorophyll_calculator_result"
LIGHT_NAME = "optics_etot_ndcy"

class ChlorophyllUptoOpticalDepth(eatpy.Plugin):
    def initialize(self, variables: Mapping[str, Any], ensemble_size: int):
        # Get references to variables with chlorophyll, light, layer thickness
        self.chl = variables[CHL_NAME]
        self.light = variables[LIGHT_NAME]
        self.h = variables["h"]

        # Add a new variable for chlorophyll averaged over the first optical depth
        variables[CHL_NAME + "_1OD"] = self.chl_sf = {
            "long_name": "chlorophyll averaged over 1st optical depth",
            "units": self.chl["units"],
            "length": 1,
        }

    def before_analysis(self, *args, **kwargs):
        # Obtain model values for chlorophyll, light, layer thickness.
        # All three have shape (nensemble, nlayer)
        chl = self.chl["data"]
        light = self.light["data"]
        h = self.h["data"]

        # Select only layers with light exceeding 1/e of surface value,
        # as representative for water-leaving irradiance (https://doi.org/10.1364/ao.14.000413)
        # Average chlorophyll over these layers, accounting for variable layer thickness
        select = light > np.exp(-1.0) * light[:, -1:]
        chl_int = (chl * h).sum(axis=1, where=select)
        h_int = h.sum(axis=1, where=select)
        self.chl_sf["data"] = chl_int / h_int
```

**Figure 5. Example of an EAT plug-in, written in Python, that implements an observation operator that calculates the mean chlorophyll concentration over the first optical depth in the model. The plugin augments the model state with the custom chlorophyll diagnostic. This diagnostic can subsequently be linked to observations (e.g., Fig 3). The calculation is implemented in the plugin's "before_analysis" method, which is called just before model state and observations are sent to the filter (PDAF) for the data assimilation update.**

Most real-world data assimilation experiments require user-specified logic as part of the data assimilation update, e.g., to implement custom operators that reconstruct observation equivalents from the model state, to limit the data assimilation update to a subset of the model state, to apply anamorphosis functions that transform variables into a "Gaussian" space, to apply additional constraints to state variable values (e.g., to ensure values remain physically meaningful, or to ensure mass conservation), or to specify the background error covariance matrix in variational schemes. EAT makes this possible through *plugins*: snippets of Python code that execute during initialization, just before the assimilation update, and just after the assimilation update. These plugins have read/write access to the model state variables and all assimilated observations, which allows them to a variety of things: during initialization, they can add/remove variables to the state seen by the DA filter, at runtime they can transform state variables and/or observations, check and log variable ranges and clip values, or save any element of the assimilation update (forecast, analysis, observations) to file. EAT includes example plugins that perform each of these functions; users can also implement their own, typically directly within the run script. An example of a user plugin implementing a custom observation operator is shown in Fig. 5. EAT's plugin infrastructure is additionally used with variational data assimilation schemes (parameterized and hybrid 3D-Var) to allow the user to provide custom routines for covariance transformation; these are then called during the iterative state update. Any number of plugins can be active during an EAT experiment; they will be called sequentially. EAT's flexibility extends beyond plugins: users can also implement custom data assimilation

filters and use these instead of the provided PDAF options; this requires just one Python function that takes in the forecast state and observations and returns the analysis state.

## 3. Applications

The EAT package has been applied at different locations in the European seas in three different configurations (table 1): coupled physical-biogeochemical state estimation by ensemble-based assimilation, biogeochemical
state estimation by variational assimilation, and joint state-parameter estimation by ensemble-based assimilation. All three setups used meteorological forcings extracted from ERA5 (Hersbach et al., 2023). The model setups and EAT scripts are publicly available (see data availability section).

**Table 1. Summary of the EAT applications at three locations with different assimilation setups.**

| Assimilation method | Location | Model | Assimilated observations | State estimation | Parameter estimation |
|---|---|---|---|---|---|
| sequential ensemble-based (ESTKF) | North Sea | PISCES | satellite sea surface temperature and chlorophyll | temperature, salinity, all biogeochemical variables | - |
| variational (3D-Var) | Mediterranean Sea | BFM | satellite chlorophyll | 17 phytoplankton variables | - |
| sequential ensemble-based (ESTKF) | English Channel | ERSEM | North West European Shelf-sea reanalysis assimilating OC PFT chlorophyll | 5 diatom variables | maximum specific productivity at reference temperature for diatoms |

### 3.1.      Physical-biogeochemical state estimation with ensemble-based assimilation

Different types of coupled physical-biogeochemical data assimilation were tested for a 110 m deep site in the Northern North Sea (59.33° N, 1.28° E) using the FABM implementation of the PISCES biogeochemical model (Aumont et al., 2015). The model is set up to cover the period 2020-2022. Initial conditions were taken from the World Ocean Atlas 2018 (temperature, salinity, nitrate, phosphate, silicate, oxygen) and GLODAP v2 (total
dissolved inorganic carbon, alkalinity). Meteorological forcing was taken from the ERA5 reanalysis. Tidal forcing was implemented by prescribing horizontal gradients in surface elevation, taken from TPXO9 (Egbert and Erofeeva, 2002).

Two data assimilation experiments were performed: the first assimilated only daily sea surface temperature observations from the level 4 SST-CCI product, the second additionally assimilated surface chlorophyll
observations from the level 3 OceanColour CCI product. Both types of observations were mapped to model equivalents in the very top layer of the modelled water column, which is 10 cm thick. In the first experiment, the assimilation scheme operates on temperature and salinity only (weak coupling: biogeochemical fields are affected through simulation with the coupled model system). In the second experiment, the assimilation scheme operates on temperature, salinity, all biogeochemical state variables defined by the PISCES model, and the
diagnostic total chlorophyll concentration, obtained by summing chlorophyll in PISCES' two phytoplankton function types (PFTs) (strong coupling: assimilation is applied simultaneously to the full system state, with covariances between physical and biogeochemical components allowed to be non-zero). All biogeochemical variables were log-transformed to guarantee positivity, as common in biogeochemical data assimilation (e.g., Santana-Falcón et al., 2020; Skákala et al., 2022; Pradhan et al., 2020). Transformation was done with a
standard plugin provided by EAT (the Log plugin in Fig. 3), which applies log-transformation to a user-specified subset of model variables and by default also to any associated observations. For the latter, the mean and variance of the log-transformed observation is reconstructed from the untransformed mean and variance under the assumption that each observation is perfectly log-normally distributed.

This application uses ensemble-based sequential data assimilation with the Error Subspace Transform Kalman
filter (Nerger et al., 2012), using an ensemble of 20 members. The initial ensemble was generated by applying log-normally distributed scale factors to:

- the x- and y-components of the surface wind (u10, v10)
- the minimum turbulent kinetic energy ($k_{min}$), which in GOTM is typically used to parameterise unresolved processes that contribute to vertical mixing (e.g., internal waves)
- the maximum growth rate of nanophytoplankton and diatoms


This was implemented using EAT's built-in support for ensemble generation (Fig 2). The above parameters were selected to showcase the ability to introduce uncertainty in meteorological forcing (u10, v10) as well as the parametrization of the physical and biogeochemical models. Incorporation of other sources of uncertainty in the ensemble will be discussed at the end of this section. All ensemble members start from the same initial conditions; spread in the ensemble state first seen by the DA filter is generated by simulating 12 hours to the time of the first SST observation.


For simplicity, the same probability distribution was used to scale all five parameters: scale factors were drawn from a log-normal distribution with a standard deviation of 0.2 in natural log units. This was done independently per ensemble member, for each of the five scale factors (i.e., they are independently distributed). The assumption of log-normality is common in biogeochemistry (e.g., Campbell, 1995) and a reasonable choice for many turbulent quantities (H. Burchard, pers. comm., 2024). It also ensures that the affected variables remain positive definite; we note, however, that both the type of distribution and its parameters are easy to customize (Fig. 2).


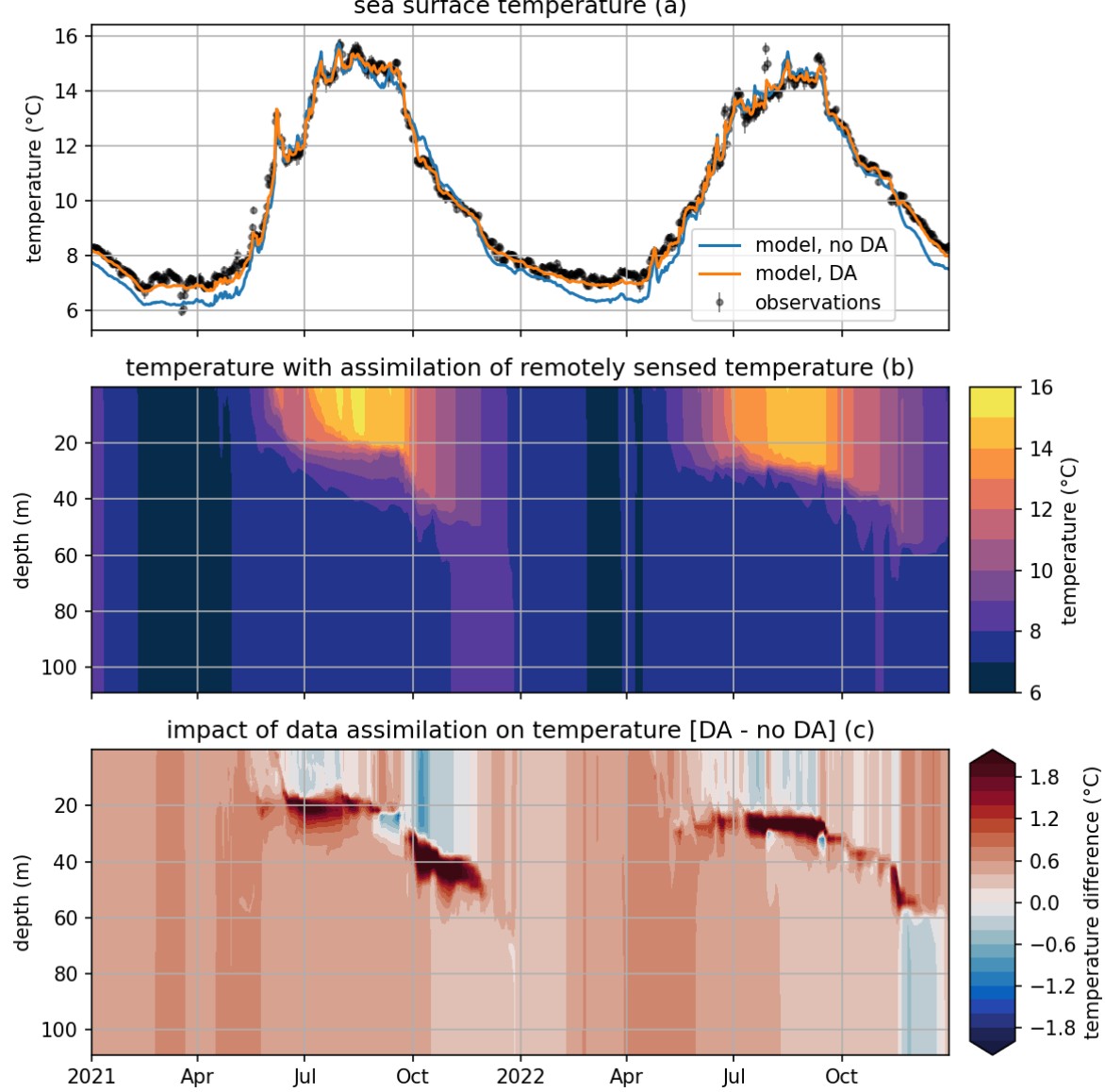

**Figure 6. Modelled and observed temperature in a free model run ("no DA") and in the experiment where remotely sensed sea surface temperatures were assimilated ("DA"). The top panel (a) shows surface temperatures (modelled and observed), the middle panel (b) shows the temperature throughout the water column in the free run, and the bottom panel (c) shows the difference in temperature between the assimilation experiment and the free run.**

Figure 6a shows the impact of data assimilation on sea surface temperature (SST). The free-running model can
be seen to have a cold bias in autumn and winter, which is eliminated when SST is assimilated. Figure 6c shows the impact of assimilation on temperature throughout the water column, compared to the free-running reference of Figure 6b. It can be seen that data assimilation already has a pronounced impact in summer, causing slightly colder surface temperature and considerably warmer (>1°C) temperatures below the original thermocline, suggesting enhanced vertical mixing. This pattern generally persists into autumn, although occasionally, warmer
surface temperatures in the DA experiment cause a decrease in mixing, and accordingly a shoaling in thermocline depth that manifest as subsurface cooling.

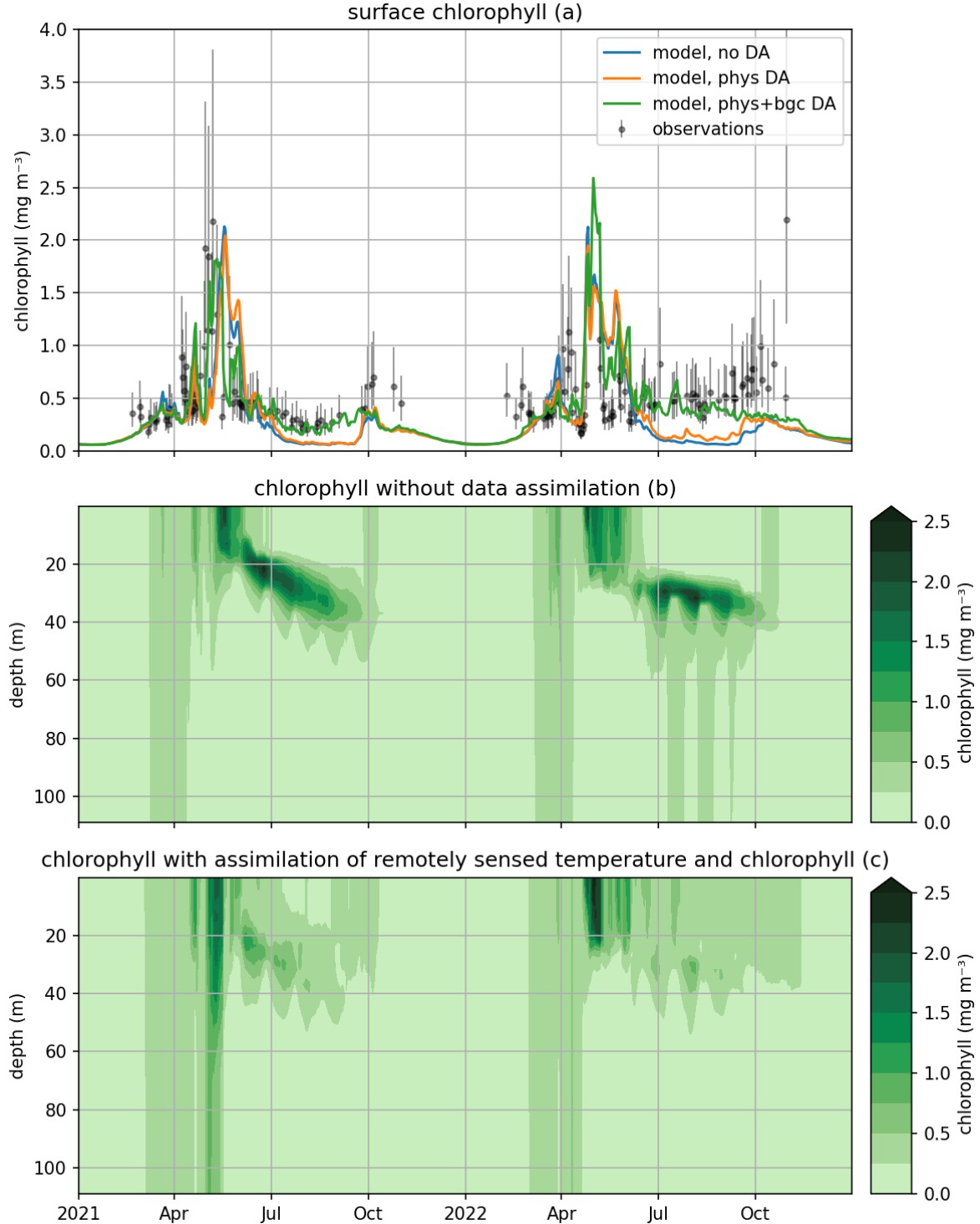

**Figure 7. Modelled and observed total chlorophyll in a free model run ("no DA"), the experiment where remotely sensed sea surface temperatures were assimilated ("phys DA"), and the experiment where remotely sensed temperature and chlorophyll were assimilated ("phys+bgc DA"). The top panel (a) shows surface chlorophyll (modelled and observed), the middle panel (b) shows chlorophyll throughout the water column for the "no DA" run and the bottom panel (c) shows chlorophyll throughout the water column for the "phys+bgc DA" experiment.**

Figure 7 shows the impact of assimilation on surface chlorophyll, both through SST-only assimilation in the first experiment (weak coupling between physics and biogeochemistry, "phys DA"), and through SST and chlorophyll assimilation in the second experiment ("phys + bgc DA"). It can be seen that SST assimilation alone has limited impact on chlorophyll: its main effect is a slight increase in chlorophyll in summer (likely though the enhanced mixing caused by SST assimilation), which is beneficial but not sufficient to raise chlorophyll to

observed values. Unsurprisingly, the combined assimilation of SST and surface chlorophyll results in the model closely tracking observations throughout the simulation. The impact of assimilation is felt more strongly at depth, however: the subsurface chlorophyll maximum is much less pronounced, with concentrations dropping at least twofold.

This application can be extended in several ways. For example, additional sources of uncertainty can be introduced when constructing the ensemble. The current setup includes a primitive parametrization of meteorological uncertainty through scaling of the surface wind components; more realistic experiments might source an ensemble of different meteorological model realizations (i.e., separate meteorological forcing files) and distribute those over the EAT ensemble members. EAT facilitates this by allowing its ensemble generators to set YAML parameters (e.g., the location of meteorological forcing in gotm.yaml) to member-specific file paths, similar to how the biogeochemical configuration (`fabm/yaml_file`) is treated in Fig. 2. Another option is to introduce uncertainty in biogeochemical parameters other than phytoplankton maximum growth rate. This is easy to realize: all biogeochemical parameters are set in fabm.yaml and any of these can be varied across the ensemble by adding a single line in the ensemble generation script as in Fig. 2. Finally, it is possible to vary physical and biogeochemical initial conditions across the ensemble, as shown in the last section of Fig. 2.

Another possible extension is to assimilate observations that describe not just the water surface, but also deeper layers. Notably, the inclusion of depth-explicit biogeochemical observations, e.g., from ship-based casts, automatic profilers or Argo floats, might help determine whether the decrease in subsurface chlorophyll in Fig. 7 is realistic or an artifact of surface-only chlorophyll assimilation. Inclusion of depth-explicit observations in EAT is straightforward, as it merely requires adding a column with depth information to the observation file and dropping the depth index (-1) in the linked model variable (Fig. 3). Another option would be to account for the fact that remotely sensed chlorophyll observations may be representative for a greater depth range than just the very first model layer, which here is just 10 cm thick (Gordon and McCluney, 1975). For instance, Fig. 5 demonstrates how EAT's plugin architecture can be used to define a custom observation operator that calculates the average chlorophyll concentration over the first optical depth in the model, which varies both over time and between ensemble members (as these differ in chlorophyll-derived light attenuation). By linking remotely sensed chlorophyll to this custom chlorophyll metric, biogeochemistry at depth will be better constrained.

Finally, EAT lends itself well for further experiments that investigate the impact of assimilation of different observations. For instance, an experiment assimilating only sea surface chlorophyll (included in the application notebook; see Data Availability section) could help ascertain whether coupled physical-biogeochemical assimilation performs better or worse than biogeochemistry-only assimilation.

### 3.2. Biogeochemical state estimation with variational assimilation

The variational assimilation implemented in EAT has been tested in the Mediterranean Sea using the BFM biogeochemical model. For this purpose, GOTM-FABM has been set up for a location the Tyrrhenian Sea (12.36° E, 39.36° N). The atmospheric forcings and profiles of temperature and salinity used for nudging have been obtained using the iGOTM tool (https://igotm.bolding-bruggeman.com/). A relatively weak nudging to the temperature and salinity profiles has been imposed applying a one-year relaxation time.

The BFM model describes the marine lower trophic web through the spatial and temporal evolution of 51 state variables. BFM uses variable stoichiometry and represents cycles of carbon, nitrogen, phosphorus, silicon. Accordingly, it explicitly tracks fluxes of these elements between its nutrient pools (nitrate, phosphate and silicate) and living functional types (phytoplankton, zooplankton and bacteria) (Vichi et al., 2020).

Here a test with variational assimilation of satellite chlorophyll observations is presented. The variational assimilation is performed using an opportunely developed EAT plug-in that reflects the 3D variational scheme that is operationally implemented to provide the Mediterranean Sea biogeochemical products in the Copernicus Marine Service. In the present 1D application, the background error covariance matrix (that propagates the assimilation increments across space and variables) is composed by two elements: the vertical covariance operator and the biogeochemical one. As in the 3D application (Teruzzi et al., 2021a), the vertical covariance is described by a set of precomputed covariances (empirical orthogonal functions of a multiannual simulation), while the biogeochemical covariance is limited to the phytoplankton variables and aims at preserving the phytoplankton physiological state by keeping constant the ratio among the phytoplankton components.

In the EAT framework, we performed two simulations for 2019 using BFM to evaluate the variational assimilation and its impacts: a run with weekly assimilation of satellite chlorophyll concentration and a free run without assimilation. The simulations started from vertically varying conditions extracted from Copernicus Marine Service reanalysis (Teruzzi et al., 2021b). The assimilation is performed once a week in 2019 using weekly averaged observations at a location in the proximity of a BGC-Argo float location (12.36 °E, 39.35° W). Chlorophyll satellite observations are extracted from the near real time ocean colour (OC TAC) daily product in the Copernicus Marine Service catalogue. The comparison of the two simulations shows the assimilation effects on chlorophyll (Fig. 8) and primary production (Fig. 9). As expected, at surface the assimilation brings chlorophyll concentrations closer to satellite observations (Fig. 8a). Considering chlorophyll concentration in the water column, greater assimilation impacts occur during late winter and spring (Fig. 8c), with assimilation increasing chlorophyll concentration nearly uniformly in the 0-100 m layer in the late winter mixed bloom (from mid-January to mid-February) while decreasing it through March. At the beginning of the stratification phase (since April) effects of data assimilation are no more uniform along the water column with impacts on the localization of the subsurface chlorophyll maximum (that is visible in Fig. 8b). Most relevant effects on primary production also occur during late winter and spring (Fig. 9b), impacting the productivity of the surface winter bloom time window (Fig. 9a). Even if limited to a shallower layer (0-40 m), variational assimilation impacts on primary production are similar to those on chlorophyll in the same season, with increase of primary production between half of January and half of February and strong decrease in March. Slighter effects on primary production occur in summer and in the autumn transition phase.

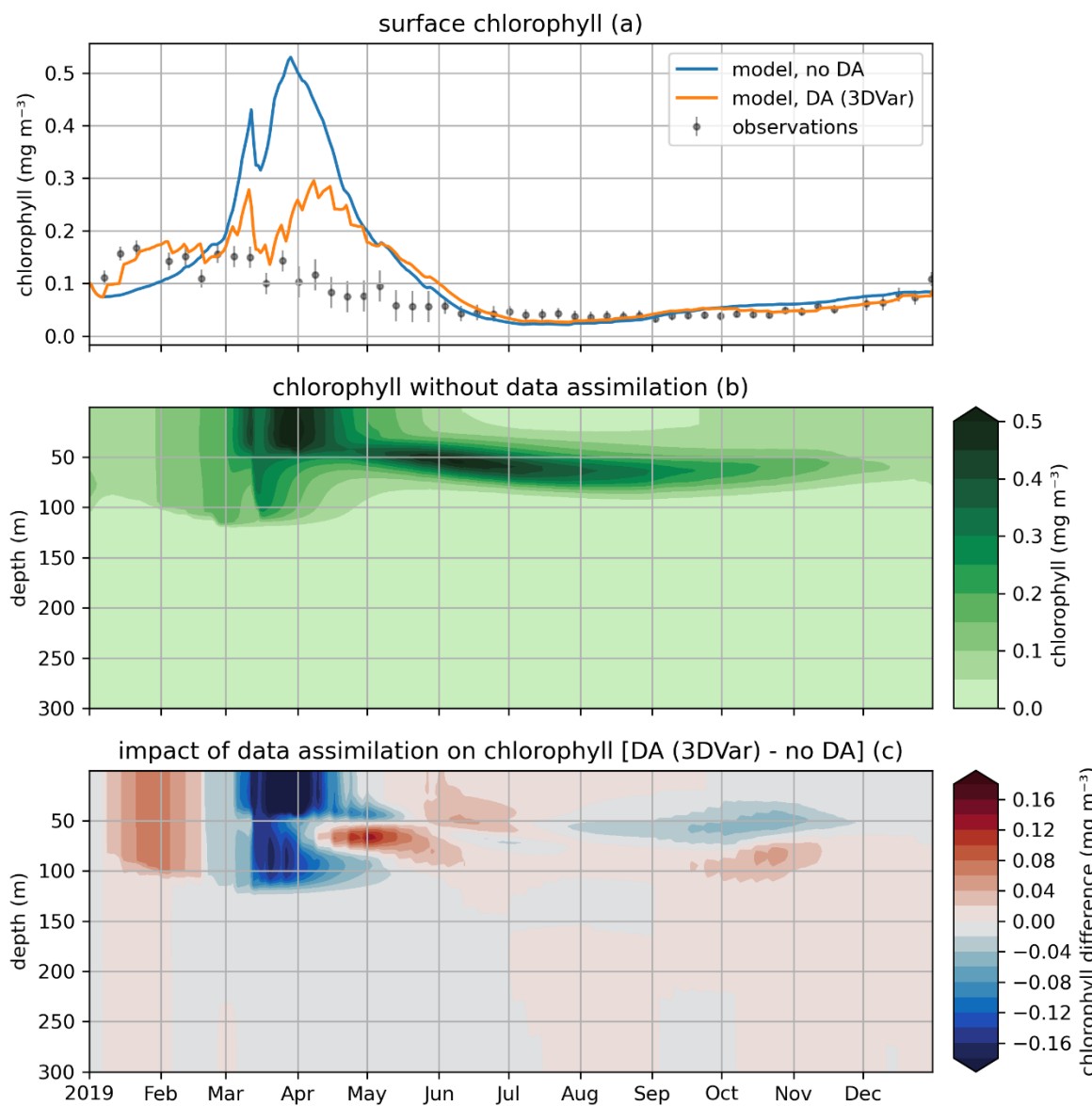

**Figure 8.** Modelled and observed chlorophyll in a free model run ("no DA") and in the experiment where remotely sensed surface chlorophyll were assimilated ("DA (3DVar)"). The top panel (a) shows surface chlorophyll (modelled and observed), the middle panel (b) shows the chlorophyll throughout the water column in the free run, and the bottom panel (c) shows the difference in chlorophyll between the variational assimilation experiment and the free run.

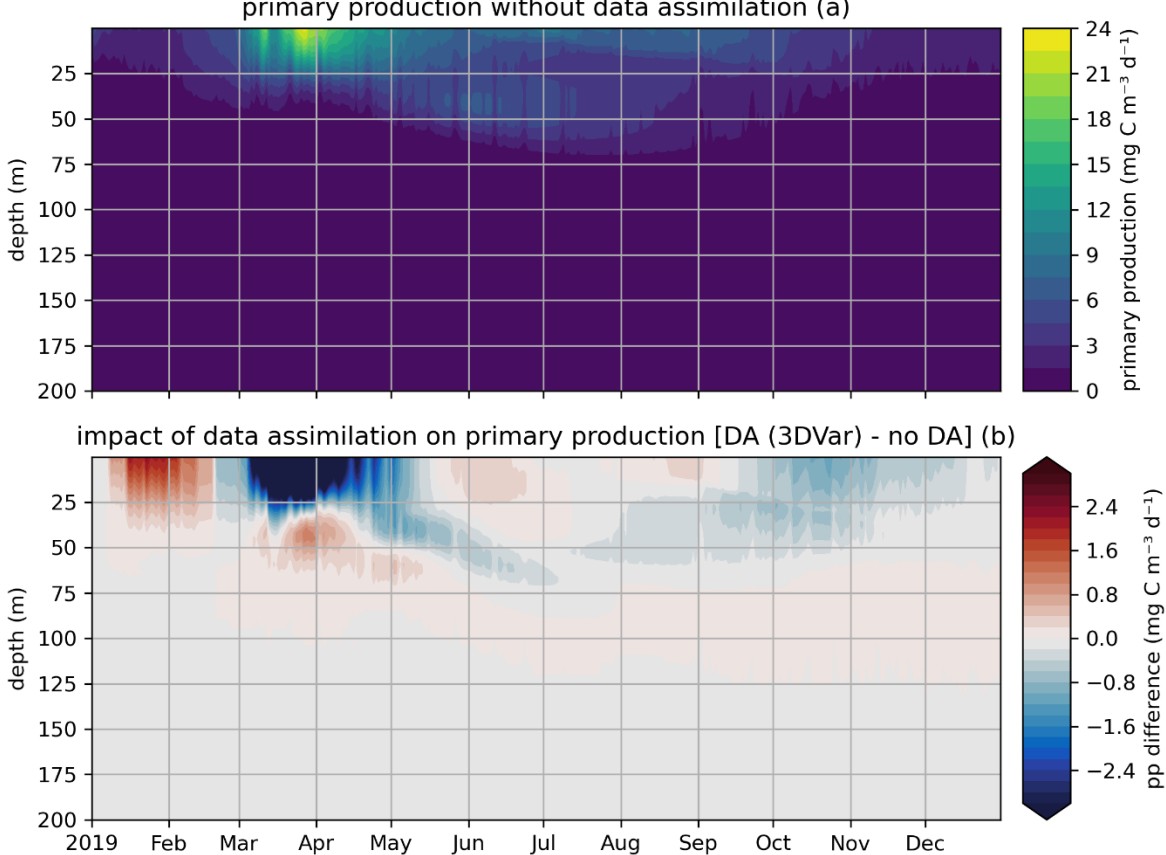

**Figure 9. Modelled primary production in a free model run ("no DA") and in the experiment where remotely sensed surface chlorophyll were assimilated ("DA (3DVar)"). The top panel (a) shows the chlorophyll throughout the water column in the free run, and the bottom panel (b) shows the difference in chlorophyll between the variational assimilation experiment and the free run.**

This application can be easily extended. To better constrain the behavior of the model at depth, it would be possible to include chlorophyll profiles as part of the assimilation, or to use a custom observation operator that considers a greater depth range when calculating the model equivalent of remotely sensed chlorophyll. Both options are described in more detail for the previous application. Further, the background error covariances used in the variational approach can be reformulated to extend the assimilation to variables other than chlorophyll (e.g., profiles of nitrate or oxygen). This is straightforward to implement, as the background error covariances in EAT are already controlled by a user-defined plugin when using variational assimilation. A combination of plugins can be used to fully specify how biogeochemical variables are affected by the assimilation: to control which model variables are of interest for assimilation (the Select plugin in Fig 3), to read and apply predefined vertical error covariances, and potentially to implement custom schemes that adjust filter-proposed updates or extend them to others variables, e.g., by applying preservation rules that force ratios between selected model variables to remain the same before and after DA.

### 3.3. Parameter estimation

We demonstrate the use of EAT software in an experiment where uncertain biogeochemical parameters are estimated in time-dependent way with the use of ensemble data assimilation techniques established in PDAF. The EAT system was run at the L4 location, which is an observing station in the western English Channel (part of Western English Channel Observatory, https://www.westernchannelobservatory.org.uk/), within the near-coastal zone, around 15 km from the Plymouth Sound (50° 15.00' N, 4° 13.02' W). The location is 50 m deep and is characterized by seasonally stratified dynamics (Pingree and Griffiths, 1978), with significant input from nearby river mouths (e.g., Tamar, Plym rivers). The observing station at L4 provides data for essential physical variables (e.g., temperature, salinity) and one of the longest continuous time-series (since 1988, Harris, 2010) for a number of biogeochemical variables, such as total phytoplankton chlorophyll and carbon biomass, nutrients (nitrate, phosphate, silicate, ammonium) and oxygen.

We have focused on the European Regional Seas Ecosystem Model (ERSEM), a highly complex biogeochemistry model with > 50 pelagic state variables and > 400 model parameters (Butenschön et al., 2016), most of which are highly uncertain. ERSEM uses variable stoichiometry, representing cycles of multiple chemical elements (carbon, nitrogen, phosphorus, silicon), with four functional types of phytoplankton (picophytoplankton, nanophytoplankton, microphytoplankton, diatoms) and three functional types of zooplankton. From the many (mostly poorly constrained) ERSEM parameters we selected in this experiment the maximum specific productivity at reference temperature of diatoms (Butenschön et al., 2016), which has been identified as one of the 5 most sensitive ERSEM parameters from the point of simulating a selected class of ecosystem target indicators (Ciavatta et al., 2022). In the following, we will refer to this parameter as "diat-MSP". The GOTM-FABM-ERSEM 1D configuration was run for the 3-year period between November 2014 and October 2017, using initial conditions from a 7-year spin-up run.

In our experiment we used a 50-member ensemble originating purely from the estimated diat-MSP parameter perturbations, where the initial prior perturbations were drawn from a uniform distribution with ±30% interval around the currently used parameter value (diat-MSP=1.375 $d^{-1}$). Although by limiting ensembles only to perturbations of the diat-MSP parameter we lack realistic representation of the background uncertainty, it is still the most pragmatic choice given the constrained size of the ensemble, since introducing additional perturbations would introduce significant noise into the parameter-state cross-covariances. On the other hand, using only diat-MSP parameter perturbations for the ensemble, introduces perfect correlation between the parameter and the assimilated variable, potentially introducing spurious short-term fluctuations into the diat-MSP values. To remove this effect, we have low-pass filtered the diat-MSP time-series on a scale of a month.

The EAT assimilation method chosen for our experiment was ensemble-3DVAR relying on ESTKF for ensemble transformation. The optimal data for assimilation are diatom chlorophyll, which were not available among the L4 data for the relevant period. We thus decided to use the data from the North-West European Shelf bi-decadal reanalysis (https://doi.org/10.48670/moi-00059, Skákala et al., 2023) produced by the UK met Office, assimilating chlorophyll derived from the ocean color satellite measurements, and partitioned into phytoplankton functional types (including diatoms) (Brewin et al., 2017). The reanalysis validates nicely against many observed L4 variables (Skákala et al., 2023), with the comparison of total chlorophyll being slightly worse, probably due to representativity issues and noise in L4 observations (see some discussion in Skákala et al., 2023). The reanalysis data were assimilated into the model every 5 days, updating all the ERSEM diatom biomass components and the estimated diat-MSP parameter.

Figure 10 shows the time-series for diat-MSP, demonstrating that the parameter is highly time-variable. This suggests that, as far as model performance is concerned, it is better to change the model structure by accounting for diat-MSP time-variability, rather than fitting time-constant parameters as it is done in the present ERSEM. Such temporal (as well as spatial) variability in diatom parameters could account for changes in the internal diatom species composition, which remains unresolved by the ERSEM model, suggesting that ERSEM needs improving in its capability to capture biodiversity. However, we need to keep in mind that Fig. 10 is only a temporally varying parameter fit, and as such should not be over-interpreted, e.g., the varying parameter values could be just a simple bias-compensation for various other model deficiencies, which could be also potentially related to the large numbers of other poorly constrained ERSEM parameters.

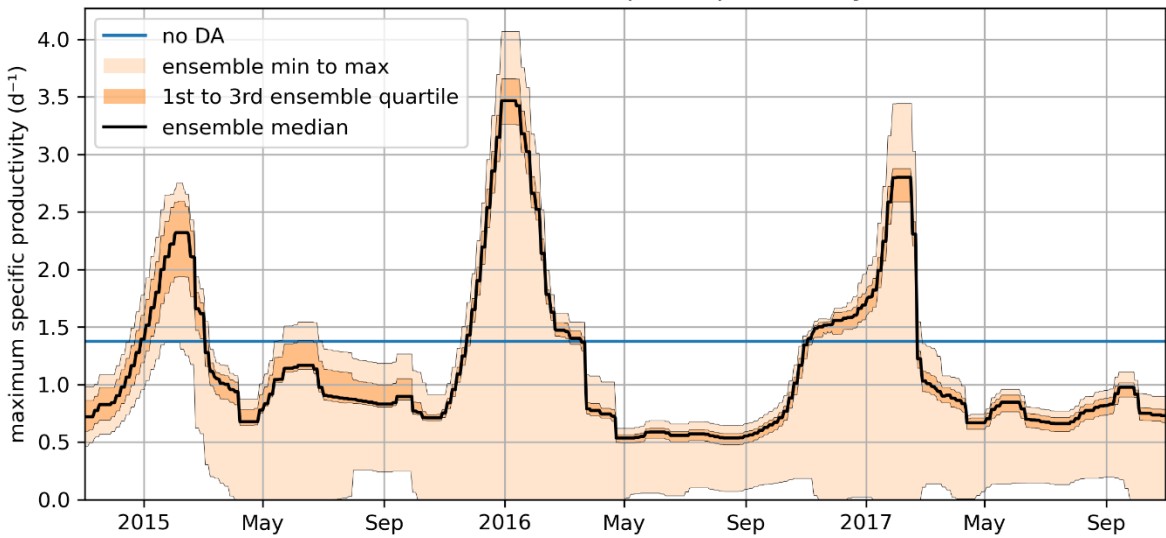

**Figure 10. The November 2014 – October 2017 time series for the diat-MSP parameter, showing the median value, the ensemble spread and the region between the two quartiles around the median. The figure also shows the (constant) parameter value in the non-assimilated case (blue line).**

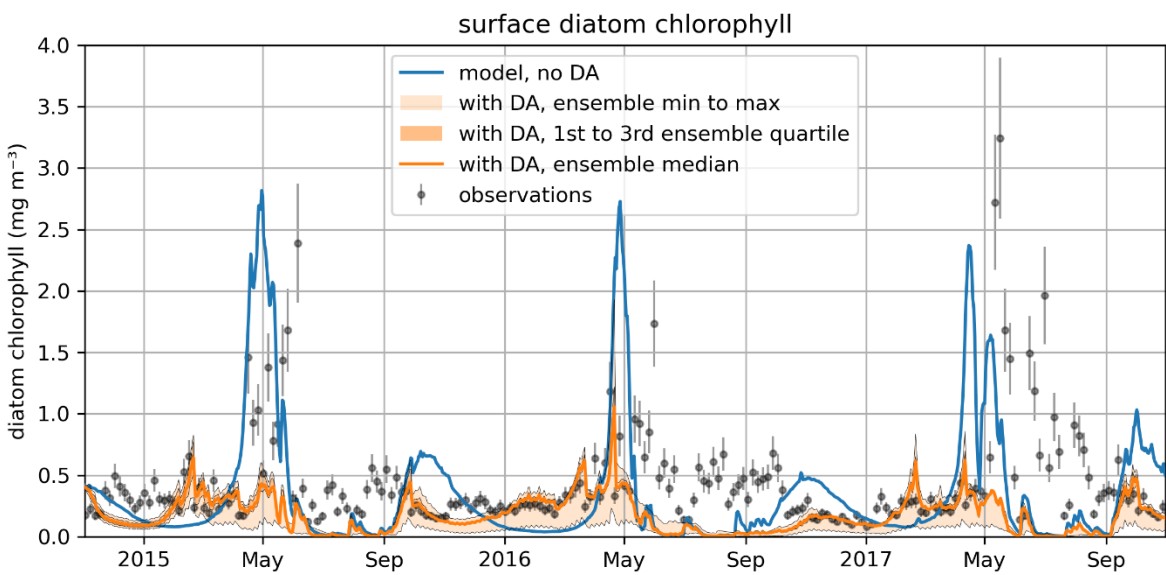

**Figure 11. The time-series for surface diatom chlorophyll-a concentrations comparing the median of the model free run ensemble, the assimilated observations from reanalysis, and the median of the ensemble with data assimilation active.**

Figure 10 also shows that the time-variability of diat-MSP is dominated by the strong seasonal harmonics, with high parameter values during Winter and low parameter values during Summer. It is then understandable that the diat-MSP parameter shows correlation with temperature (R=-0.55) and also with chlorophyll (R=0.48). The diat-MSP parameter seasonal variations can be understood from the Fig. 11, showing that the diat-MSP values compensate for the model seasonal biases in the diatom concentrations, i.e. both for the model diatom underestimates in the Winter and their overestimates during the Spring bloom. Fig. 11 also indicates that the Summer low diat-MSP values from Fig. 10 might be for a large part a relic of low ensemble spread, leading to the lack of assimilation impact in the Summer (no assimilation means that the parameters will retain their last acquired values).

This application could be extended to estimate biogeochemical parameters beyond the maximum growth rate of diatoms. From a technical point of view, this is straightforward: the list of biogeochemical parameters that are to be estimated is configured in the run script through argument `fabm_parameters_in_state` provided to `eatpy.models.GOTM` (for the application's original run script, see Data availability section). The (scientific) challenge is to decide which specific parameters to target, as some biogeochemical models have several hundreds of parameters. One potential strategy is to first perform sensitivity analysis to determine which biogeochemical parameter have the greatest impact on model results; tools for doing this with GOTM-FABM are readily available (Ciavatta et al., 2022; Andersen et al., 2021). Another important consideration for selecting parameters is to assess which biogeochemical parameters are most likely to exhibit temporal variability. That assessment should highlight parameters that are constant in the original model, but clearly aggregated over multiple processes or functional types, and thus likely to vary temporally in reality as the relative importance of their constituent processes or functional types changes. Finally, it is worth noting that in EAT, parameters that are estimated during assimilation remain the same over the entire water column, even though they become variable in time. This is also common in 3D data assimilation (Doron et al., 2013). However, if parameter variability is assumed to stem from shifts in biological species composition, it is worth noting that the current approach cannot account for e.g. separate "light" and "shade" communities (Sournia, 1982), which would require parameter values to vary in depth. This can only be achieved by modifying the (Fortran) source code of the biogeochemical model in order to replace each affected (scalar) parameter by a spatially resolved state variable.

## 4. Discussion

The EAT software package has several features that make it well suited for exploring the latest developments in marine data assimilation. First, its core 1D model, GOTM-FABM, is an *online coupled* hydrodynamic-biogeochemical model: it simultaneously simulates the physical and biogeochemical state of the water column. This combined state is available to the data assimilation filter, which means that (1) observations of any physical or biogeochemical variable can be assimilated, and (2) during an assimilation update, observational information can propagate to any part of the physical-biogeochemical state via emergent or prescribed cross-covariances. This enables *coupled* data assimilation, for instance, experiments that assess how assimilation of physical observations affects modelled biogeochemistry, or vice versa. Such coupling can either be "weak" (assimilated observational information propagates to other variables during simulation with the coupled model system) or "strong" (observational information additionally propagates to multiple variables during assimilation updates via cross-covariances) (Penny et al., 2017) (see the "Coupled physical-biogeochemical data assimilation" application). The propagation of information from biogeochemistry to physics placed further demands on the model system: first, it requires online physics as in GOTM-FABM, as opposed to parametrized physics (Pelc et al., 2012; Eknes and Evensen, 2002; Simon and Bertino, 2012; Bertino et al., 2003) or offline physics (Lenartz et al., 2007). Second, it benefits from a model system such as GOTM-FABM that explicitly represents feedbacks from biogeochemistry to physics, for instance, light absorption by BGC variables that heats the water, thereby changing density stratification and (turbulent) mixing (Skákala et al., 2020). These feedbacks modulate the link between BGC and physics; in weakly coupled DA experiments, they are the only mechanism through which biogeochemistry can influence physics.

A second key feature of EAT is that the biogeochemical state available to the assimilation system is readily extensible. Unlike other studies (Torres et al., 2006), it already includes benthic state variables, which permits benthic observations to be assimilated and to study the coupling between pelagic and benthic systems in data assimilation. In addition, the model state can be augmented with biogeochemical diagnostics, for instance, process rates such as net primary production (Mamnun et al., 2022). Any diagnostic already exposed by the biogeochemical model is available for this purpose; there is no need to reimplement it as part of an observation operator within EAT. Even simple expressions dependent on model state, such as the sum of chlorophyll over multiple plankton functional types, are typically already available as BGC diagnostics and therefore do not require custom user code in order to be assimilated. Finally, the model state can be augmented with any biogeochemical parameter to perform parameter-state estimation (see the "Parameter estimation" example). The value of such parameters then changes over time in response to observations being assimilated (Gharamti et al., 2017b, a; Simon et al., 2015). At present, temporal but not vertical variation in these parameters is considered: at any given time, the parameter value is the same across the water column. This, however, is already sufficient to mimic experiments that consider temporal and horizontal variation in parameters, as for instance in Doron et al. (2013).

With respect to the data assimilation methodology, EAT provides a wide choice of methods, covering both variational data assimilation and sequential estimation methods that include different ensemble Kalman filter variants. This allows users to assess the impact of different assimilation methods under identical modelling conditions. Moreover, the effect of different configuration options or representations of the covariance matrix in parameterized variational methods can be easily examined. Furthermore, EAT supports hybrid approaches that combine variational and ensemble assimilation. In this case, the user plugin for control variable transforms (an example can be found in the variational application; see Data availability section) gets access to the ensemble state, which allows the user to combine the background-error covariance matrix with the ensemble-based one in a variety of ways (Bannister, 2017). Lastly, EAT allows to perform twin experiments in which synthetic observations are assimilated and the true state, which is to be estimated, is known. For this, a simple model run with EAT can produce the true state. Synthetic observations are subsequently generated by sampling this true state and adding perturbations. Starting from a different initial state, one can then assimilate these observations. This approach allows one to study what the application of data assimilation can achieve in an optimal case. Likewise, it can give an indication of the ability of certain observations to constrain the model state or parameters. In general, EAT is flexible: user plugins are given full control over observations, forecasts, and analyses (with the ability to override proposed state updates). Ensemble members can differ in both state and configuration, and ensemble states can be saved and reused through support for restart files. These features can be combined in any number of ways to design new data assimilation experiments. Thus, the applications described here are representative of EAT functionality, but not exhaustive.

Finally, by building on established frameworks, EAT can offer the same state-of-the-art process descriptions and data assimilation algorithms that are used in operational data assimilation systems. Through GOTM, it supports a comprehensive library of turbulence closure schemes, as well as empirical vertical mixing schemes such as KPP (Li et al., 2021). Through FABM, EAT has access to a large and rapidly growing collection of biogeochemical models, including many used in reanalysis, forecasting and climate studies (e.g., ERSEM, BFM, PISCES, MEDUSA, ECOSMO, ERGOM). Through PDAF, it has access to the latest data assimilation algorithms, including ensemble-based, variational (3D-Var), and hybrid methods.

1D data assimilation systems such as EAT are valuable on their own for research and operational use (Thomas et al., 2020), but they also often serve as stepping stone to incorporating new DA theory and methods in 3D operational systems. EAT facilitates this by building on GOTM, FABM and PDAF. These frameworks and their underlying models and algorithms are widely used in existing 3D data assimilation systems. Therefore, knowledge gleaned through EAT about optimal strategies for coupled data assimilation, parameter evolution, and data assimilation methods can transfer readily to 3D.

Nevertheless, 1D models behave differently from 3D models in some respects. Their physics tend not to exhibit the (bounded) chaotic behavior associated with 3D models (Carrassi et al., 2018), and therefore, they do not show the same sensitivity to initial conditions. For instance, a 1D water column model set up for shallow sites is often fully mixed in winter, with the water temperature converging to the temperature of the overlaying air. At that moment, any initial variations in water temperature across any ensemble disappear. If ensemble members differ *only* in water temperature, its spread then collapses entirely, causing ensemble methods to fail. 1D data assimilation therefore depends on additional methods for generating ensemble spread, e.g., by perturbing forcing or parameters of physical and biogeochemical processes. EAT includes flexible ensemble perturbation logic specifically for this purpose. While this is crucial for 1D applications, it can also be helpful to explore alternative perturbations strategies that are under consideration for 3D application. Another reason why 1D data assimilation systems such as EAT cannot be fully consistent with 3D is the additional requirement of the horizontal spatial correlation structure in 3D, and through that structure, the impact of geographically distant observations on the local model state. This is still best investigated in 3D, though we note that remote observations might crudely be represented in 1D by incrementing the observation uncertainty with an estimate of the spatial covariance, potentially derived from 3D simulation results. A related aspect that is difficult to represent in 1D is the regionalized setting of estimated parameters (Brankart et al., 2012), though some aspects of this may be investigated through the use of multiple 1D DA setups across large regions and subsequent analysis of spatial patterns in their results (Skákala et al., 2023). Finally, 1D models have limitations independent of data assimilation. As they assume horizontal gradients are negligible, they cannot represent conditions in areas dominated by horizontal features, e.g., high-energy horizontal currents (e.g., the Gulf Stream) or convection currents. Fortunately, these areas cover a minor fraction of the open and coastal oceans.

Moreover, GOTM includes mechanisms to prescribe horizontal gradients, though these cannot respond to assimilation.

## 5. Conclusions

EAT is a 1D framework for marine data assimilation with numerous advantages. It is accessible: it can be installed on any computer workstation (Windows, Linux, Mac) with 2-5 commands, and is therefore readily usable by students as well as established scientists. It is flexible: through FABM, it can integrate a wide range of third-party biogeochemical models, including ones that are not distributed with EAT/GOTM/FABM. Moreover, through EAT's plugin architecture, users can readily add custom logic for ensemble generation, variable transformation and anamorphosis, covariance transformation for variational DA, ensemble diagnostics, and bespoke output in any format. Finally, EAT includes the functionality needed to replicate and develop state-of-the-art research in marine DA: it supports fully coupled physical-biogeochemical simulation and assimilation, and, through state augmentation, it supports assimilation of observed diagnostics and estimation of biogeochemical parameters. We believe this feature set makes it ideally suitable for a wide range of applications.

## 6. Code availability

All source code is publicly available, though for most applications it suffices to install a pre-compiled EAT package from Anaconda (https://anaconda.org/conda-forge/eatpy).

The EAT source code is available from https://github.com/BoldingBruggeman/eat. It includes compatible versions of GOTM, FABM and PDAF as submodules. These individual components are also available stand-alone from https://github.com/gotm-model/code (GOTM), https://github.com/fabm-model/fabm (FABM) and https://github.com/pdaf/PDAF (PDAF).

The exact version of the combined codes that were used is available at https://doi.org/10.5281/zenodo.11111437.

EAT documentation is available at https://github.com/BoldingBruggeman/eat/wiki.

## 7. Data availability

The three example applications are available from https://doi.org/10.5281/zenodo.11111361. This archive includes model configurations, observations, forcing data, run script and pre/post processing scripts.

## 8. Author contribution

Conceptualization: JB, KB, SC. Software, methodology: JB, KB, LN. Preparation of example applications was led by AT and SS. Individual examples: JB (coupled assimilation), AT and SS (state estimation with variational assimilation), JS (parameter estimation). Funding acquisition: SC, KB. Project administration: SC. JB led the writing of the manuscript. All authors contributed to writing, reviewing and editing, and approved the final manuscript.

## 9. Competing interests

The authors declare that they have no conflict of interest.

## 10. Acknowledgements

This work was funded by the project SEAMLESS ("Services based on Ecosystem data AssiMiLation: Essential Science and Solutions"), which has received funding from the European Union's Horizon 2020 research and innovation programme under grant agreement No 101004032. Part of the work was supported by the project NECCTON ("New Copernicus Capability for Trophic Ocean Networks"), which has received funding from Horizon Europe RIA under Grant Number 101081273. We acknowledge the CINECA award under the ISCRA initiative, for the availability of high-performance computing resources and support.

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
