# Peer review of "EAT v0.9.6: a 1D testbed for physical-biogeochemical data assimilation in natural waters"

_Geoscientific Model Development, 2023_

## Author Comment (AC2)

**EAT v0.9.6: a 1D testbed for physical-biogeochemical data assimilation in natural waters**

Jorn Bruggeman, Karsten Bolding, Lars Nerger, Anna Teruzzi, Simone Spada, Jozef Skákala, and Stefano Ciavatta

RC1: 'Comment on gmd-2023-238', Anonymous Referee #1, 03 Jan 2024

Citation: https://doi.org/10.5194/gmd-2023-238-RC1

The authors present a data assimilation (DA) test bed based on a 1-dimensional physical-biogeochemical water column model. The manuscript contains 3 example applications for the DA framework, which highlights its versatility regarding different model configurations and DA techniques. It could be a useful tool for beginners to learn about DA, or for practitioners to test modifications to existing DA systems. The manuscript is well written and easy to follow. I tested one of the example applications on a computer, and it ran with just a few small issues. In the manuscript I would like to see a bit more accompanying information about the test cases, in particular, related to how easy it would be to modify some of the implementation aspects.

**We would like to thank the referee for their careful review and constructive comments. Our point-by-point replies are provided hereafter in green.**

**General comments**

Overall, the three test cases that are presented in the manuscript, and also included as example applications in the downloadable software, are very instructive and helpful to a potential user of EAT. Each of the test cases appears to highlight an issue or weakness of the chosen modelling/DA approach and made me think of possible extensions of the test cases to further investigate or mitigate those issues. Here, it would be useful to include more information in the manuscript to describe how much user input or work would be required to extend the test cases.

**We are very grateful to the referee for having run the test cases. Moreover, we appreciate the opportunity to describe some of the advanced functionality that EAT has to offer. As suggested, we will for each test case include a section that describes how it may be extended. Further details are given below.**

Case in point, in test case 1 (Section 3.1) it would be interesting to examine the inclusion of more sources of uncertainty, beside the 3 or 5 parameters that are included in the ensemble generation. I am not suggesting the authors make any changes to the test case, but it would be helpful to describe the amount of change required to include other parameters here (Fig 2 appears to suggest it requires just a small change in one of the python files) or introduce changes to the initial conditions. This additional information could be included in a final paragraph of the section.

**We fully agree and will add the following paragraphs to the first application (note that this new paragraph also includes information about subsurface chlorophyll, depth-dependent observations, and custom observation operators, all referred to in later referee comments):**

[revised manuscript text omitted]

Similarly, in test case 2, the biogeochemical covariance is limited to the phytoplankton variables. Mention how easy it would be to expand it, for example, to include nitrate.

**We will add the following paragraph at the end of the variational application:**

**"This application can be easily extended. To better constrain the behavior of the model at depth, it would be possible to include chlorophyll profiles as part of the assimilation, or to use a custom observation operator that considers a greater depth range when calculating the model equivalent of remotely sensed chlorophyll. Both options are described in more detail for the previous application. Further, the background error covariances used in the variational approach can be reformulated to extend the assimilation to variables other than chlorophyll (e.g., profiles of nitrate or oxygen). This is straightforward to implement, as the background error covariances in EAT are already controlled by a user-defined plugin when using variational assimilation. A combination of plugins can be used to fully specify how biogeochemical variables are affected by the assimilation: to control which model variables are of interest for assimilation (the Select plugin in Fig 3), to read and apply predefined vertical error**

**covariances, and potentially to implement custom schemes that adjust filter-proposed updates or extend them to others variables, e.g., by applying preservation rules that force ratios between selected model variables to remain the same before and after DA."**

Then there is a (perhaps worrying) decline in subsurface chlorophyll brought by the assimilation of surface chlorophyll. Here, EAT could be a nice test bed to evaluate modified observation operators, would this be an easy thing to implement? For example, could a chlorophyll observation be considered the sum of the top 4 grid cells? Or -- more difficult -- could the observation operator be dynamically determined based on optical depth? Again, a small paragraph suggesting changes to the test case, and a description of the effort that would be involved, would be of use for many future users of EAT.

**We expect that the referee refers to the decline in subsurface chlorophyll in Fig. 6, that is, in the first test case (North Sea + PISCES). We agree that it would be worthwhile to evaluate whether this pattern is robust if chlorophyll observations are taken to be representative for a deeper part of the water column, in particular as the top layer in the model is thin (10 cm). EAT lends itself well for such experiments, as new observation operators are easy to implement through plugins. To demonstrate this, we will add a code snippet (new Fig. 4 shown above) that implements exactly the type of observation operator that the referee proposes: it dynamically determines the first optical depth from the daily mean shortwave radiation, and then averages chlorophyll over all layers above this depth.**

In test case 3, finally, the authors suggest that perhaps more than one parameter should be included in the estimation. How difficult would it be to include more? Beginners like me do not know, but would be interested in learning more, especially if the change required is small (modification of a YAML file perhaps).

**We propose the add the following at the end of the parameter estimation application:**

**"This application could be extended to estimate biogeochemical parameters beyond the maximum growth rate of diatoms. From a technical point of view, this is straightforward: the list of biogeochemical parameters that are to be estimated is configured in the run script through argument `fabm_parameters_in_state` provided to `eatpy.models.GOTM` (for the application's original run script, see Data availability section). The (scientific) challenge is to decide which specific parameters to target, as some biogeochemical models have several hundreds of parameters. One potential strategy is to first perform sensitivity analysis to determine which biogeochemical parameter have the greatest impact on model results; tools for doing this with GOTM-FABM are readily available (Ciavatta et al., 2022; Andersen et al., 2021). Another important consideration for selecting parameters is to assess which biogeochemical parameters are most likely to exhibit temporal variability. That assessment should highlight parameters that are constant in the original model, but clearly aggregated over multiple processes or functional types, and thus likely to vary temporally in reality as the relative importance of their constituent processes or functional types changes. Finally, it is worth noting that in EAT, parameters that are estimated during assimilation remain the same over the entire water column, even though they become variable in time. This is also common in 3D data assimilation (Doron et al., 2013). However, if parameter variability is assumed to stem from shifts in biological species composition, it is worth noting that the current approach cannot account for e.g. separate "light" and "shade" communities (Sournia, 1982), which would require parameter values to vary in depth. This is beyond the scope of EAT; the relevant parameters would need to be added to the depth-explicit model state within the biogeochemical model code."**

Beyond the test cases, it might also be of interest to some to include a description of some more sophisticated features of EAT -- or a brief description of what can be implemented: For example, is there an ability to use pre-computed ensemble members for quick DA experiments without having to re-run the model? The authors further mention hybrid variational-ensemble schemes, are these included in EAT already, what kind of coding abilities are required to add a new DA technique?

**We appreciate the opportunity to further highlight the possibility to use EAT for more sophisticated DA experiments. We expect that the newly added discussion of possible extensions for each application partially addresses this. In addition, we propose to modify the third paragraph of the discussion (starting at line 500 in the submitted manuscript version) as follows (novel text in italics):**

**"**With respect to the data assimilation methodology, EAT provides a wide choice of methods, covering both variational data assimilation and sequential estimation methods that include different ensemble Kalman filter variants. This allows users to assess the impact of different assimilation methods under identical modelling conditions. Moreover, the effect of different configurations options or representations of the covariance matrix in parameterized variational methods can be easily examined. **Furthermore, EAT supports hybrid approaches that combine variational and ensemble assimilation. In this case, the user plugin for control variable transforms (an example can be found in the variational application; see Data availability section) gets access to the ensemble state, which allows the user to combine the background-error covariance matrix with the ensemble-based one in a variety of ways (Bannister, 2017).** Lastly, EAT allows to perform twin experiments in which synthetic observations are assimilated and the true state, which is to be estimated, is known. For this, a simple model run with EAT can produce the true state. Synthetic observations are subsequently generated by sampling this true state and adding perturbations. Starting from a different initial state, one can then assimilate these observations. This approach allows one to study what the application of data assimilation can achieve in an optimal case. Likewise, it can give an indication of the ability of certain observations to constrain the model state or parameters. **In general, EAT is flexible: user plugins are given full control over observations, forecasts, and analyses (with the ability to override proposed state updates). Ensemble members can differ in both state and configuration, and ensemble states can be saved and reused through support for restart files. These features can be combined in any number of ways to design new data assimilation experiments. Thus, the applications described here are representative of EAT functionality, but not exhaustive.**

As nice as it is to have a fast DA system, it is limited to a 1D (vertical water column) model. Such a model is useful, but it will not be able to serve as a test bed for many operational systems which use 3D models. As use cases for EAT, the authors mention "practical aspects such as the spatial correlation structure and regionalized setting of estimated parameters" (line 54), even though I would consider these two applications as bad examples for the use of 1D models. Examining the effect of spatial correlations of satellite chlorophyll error, for example, would require a 3D DA setup; regionalized parameter estimates would require a number of 1D models, at least one for each region (many if the region boundaries are not set). Here, it would be good if the authors acknowledged some of the limitations of 1D models in the introduction already.

**While the paragraph in question highlights ongoing developments in DA, rather than use cases of 1D systems such as EAT, we appreciate that the reader may come away with the impression that EAT can address both aspects mentioned – spatial correlations structure and regionalization of estimated parameters. We agree that this is not the case when it comes to spatial correlation structure and**

**therefore will drop this phrase. As for the "regionalized setting of estimated parameters", there is precedent for using EAT for this purpose using exactly the method that the referee describes (multiple 1D setups). Therefore, we propose to leave this part of the sentence in place. We will expand the Discussion to discuss horizontal covariances and the work done with EAT on regionalized parameter estimates:**

"Nevertheless, 1D models behave differently from 3D models in some respects. Their physics tends not to exhibit the (bounded) chaotic behavior associated with 3D models (Carrassi et al., 2018), and therefore, they do not show the same sensitivity to initial conditions. For instance, a 1D water column model set up for shallow sites is often fully mixed in winter, with the water temperature converging to the temperature of the overlaying air. At that moment, any initial variations in water temperature across any ensemble disappear. If ensemble members differ only in water temperature, its spread then collapses entirely, causing ensemble methods to fail. 1D data assimilation therefore depends on additional methods for generating ensemble spread, e.g., by perturbing forcing or parameters of physical and biogeochemical processes. EAT includes flexible ensemble perturbation logic specifically for this purpose. While this is crucial for 1D applications, it can also be helpful to explore alternative perturbations strategies that are under consideration for 3D application. **Another reason why 1D data assimilation systems such as EAT cannot be fully consistent with 3D is the additional requirement of the horizontal spatial correlation structure in 3D, and through that structure, the impact of geographically distant observations on the local model state. This is still best investigated in 3D, though we note that remote observations might crudely be represented in 1D by incrementing the observation uncertainty with an estimate of the spatial covariance, potentially derived from 3D simulation results. A related aspect that is difficult to represent in 1D is the regionalized setting of estimated parameters (Brankart et al., 2012), though some aspects of this may be investigated through the use of multiple 1D DA setups across large regions and subsequent analysis of spatial patterns in their results (Skákala et al., 2023).** Finally, 1D models have limitations independent of data assimilation. As they assume horizontal gradients are negligible, they cannot represent conditions in areas dominated by horizontal features, e.g., high-energy horizontal currents (e.g., the Gulf Stream) or convection currents. Fortunately, these areas cover a minor fraction of the open and coastal oceans. Moreover, GOTM includes mechanisms to prescribe horizontal gradients, though these cannot respond to assimilation."

Finally, I wanted to bring up the name of the tool: It is probably too late to change EAT's name, but EAT expands to "Ensemble Assimilation Tool" in my mind (it is easy to miss the "and" in "Ensemble and Assimilation Tool") even though EAT supports variational DA as well. Furthermore, it would have been nice to include "Aquatic" or "Marine" in the abbreviation, but that is just a suggestion if a name change is still on the table.

**We appreciate that the name "Ensemble and Assimilation Tool" does suffer from the minor drawbacks mentioned, but we also concur with the referee's judgment that it is too late to change this, as EAT has numerous users already (> 100 people have used it during workshops). They have written scripts that reference the "eatpy" Python package (see e.g. Fig 2 and 3), and who have "eatpy" installed as conda package. Changing the name would break their scripts and prevent them from updating the application with "conda update". We prefer to avoid this. We will however ensure that we consistently highlight its ability to do variational DA as well as ensemble-based DA.**

**Testing one of the example applications**

I decided to run test case 2 "Biogeochemical state estimation with variational assimilation" on a Linux machine. Overall, it worked well and there were only a few minor hiccups. I downloaded the zip file that was referenced in the paper, unzipped it and followed the instructions in the `README.md` file. Installing EAT using conda was straightforward and worked right away. The free run worked flawlessly as well, just the DA experiment is missing the output directory:

```
$ mpiexec -n 1 python runVar.py : -n 1 eat-gotm
INFO:root:Model simulated period: 2019-01-01 00:00:00 - 2019-12-31 18:00:00
Traceback (most recent call last):

  File "/home/user/eat-applications/Variational/BFMvar/runVar.py", line 34, in <module>
experiment.add_plugin(eatpy.plugins.output.NetCDF(outfile))
              ^^^^^^^^^^^^^^^^^^^^^^^^^^^^^^^^^^^^
  File "/home/user/.conda/envs/eat/lib/python3.12/site-packages/eatpy/plugins/output.py", line 14, in __init__
    self.nc = netCDF4.Dataset(path, "w")

          ^^^^^^^^^^^^^^^^^^^^^^^^^
  File "src/netCDF4/_netCDF4.pyx", line 2469, in netCDF4._netCDF4.Dataset.__init__
  File "src/netCDF4/_netCDF4.pyx", line 2028, in netCDF4._netCDF4._ensure_nc_success
PermissionError: [Errno 13] Permission denied: 'OUTNC/DAout.nc'
```

A quick `mkdir -p OUTNC` solved the issue, and I would recommend adding this directory to the zip (perhaps with a small placeholder file `OUTNC/README.md` which just mentions that this directory will be used for netCDF output).

Finally, I ran into some issues with the figure creation, too: `launcher.sh` was not executable and after making it (`chmod u+x launcher.sh`), I received odd error messages because the file has dos line breaks.

```
./launcher.sh:      line      2:      $'\r':      command      not      found
./launcher.sh:      line      10:      $'\r':      command      not      found
./launcher.sh:      line      14:      $'\r':      command      not      found
[...]
```

This issue might not be easily reproducible, but the script even created an odd `^M^M` directory. After changing the file format to UNIX, it all worked and created the plots. I am not sure what to do about this particular issue (will UNIX line endings cause issues on Windows machines?) but including a proper shebang `#!/bin/bash` in `launcher.sh` at least stopped the script right away without creating directories etc., so I would at least recommend that. Overall, I was impressed by how well it worked, but I did not yet attempt to make any modifications to the experiment.

**We apologize for the inconvenience and much appreciate that the referee has taken the time to work around the issues encountered.**

**All of the above issues have been addressed in a new release of the applications (v2.1 on Zenodo; https://doi.org/10.5281/zenodo.10463234). Notably, all three applications now use Jupyter Notebooks, which work on all supported platforms (Windows, Mac, Linux).**

**Specific comments:**

L 17: It is unclear what "This" is referring to, I would suggest using "DA".

**We will change the text accordingly.**

L 78: Here (and maybe already in the abstract), spell out the abbreviations used or include a reference to section 2.

**We will change the text accordingly.**

L 159: While I appreciate the discussion of the different coupling schemes and implementation details, as a new user of EAT, I would be more interested in what I need to do to run DA for my model. Case in point:

> "Online coupling is achieved by inserting function calls to PDAF. This can be done by augmenting the model source code itself, which then enables simulation of an ensemble of model states in a single execution of the model. Alternatively, for models already capable of ensemble simulations, it can be implemented in dedicated DA code after an ensemble of model results is received. The latter is the approach adopted by EAT. PDAF-specific additions to the code are usually four functions, all of which are placed outside of the actual numerical core of the model. Overall, the online coupling approach reduces the amount of data that needs to be written to files and allows efficient data assimilation, in particular when the forecast phase between two assimilation steps is short compared to the start-up time of a model, as is common for GOTM-FABM."

After reading it, I have learned some fundamentals about PDAF, but I am not sure what kind of effort is required for a typical user. Do I need to implement four functions or is more work required, are these Python or Fortran functions? I would suggest rewriting this paragraph with an emphasis on the EAT implementation and EAT-specific instructions. For example, instead of starting "There are different strategies to couple a model with PDAF. The offline coupling uses ...", one could use "While PDAF supports both offline and online coupling, EAT uses online coupling to connect the model to the DA framework..."

**We will rephrase the final part of the PDAF section to focus more specifically on its application and use within EAT:**

**"While PDAF supports both offline and online coupling (Nerger and Hiller, 2013; Nerger, 2020), EAT uses online coupling to connect the model to the DA framework: the model state is updated as part of the data assimilation step (analysis) while the simulation remains running. EAT stores the ensemble state internally in an array, which is synchronized with the active GOTM-FABM processes before and after the DA update. PDAF exposes numerous configuration options, which include the type of data assimilation filter to use as well as various filters-specific settings. EAT enables the user to set these configuration options in a Python run script. Internally, these options are then forwarded to PDAF functions."**

L 186: Is the <RUNSCRIPT> here akin to what is shown in Fig. 3? Maybe add a short explanation to the required input.

**Yes, Fig 3 is an example of such a run script. We will explicitly refer to Fig 3 and include an explanation of every input:**

**"Here, <RUNSCRIPT> is the name of the Python script that defines the data assimilation experiment (Fig. 3), <NENSEMBLE> is the number of ensemble members, and <EXTRA_ARGS> are additional arguments to pass to the model, e.g., --separate_gotm_yaml to indicate different ensemble members use different configurations, or --separate_restart_file to indicate different members use different initial states."**

Fig 2: For readability, set the mean values explicitly, even though these are identical to the default.

**We will change the code snippet accordingly (see updated Fig 2 above).**

Fig 3: I think it would be more useful to add some of the information from the caption to the code (in the form of comments).

**We will change the code snippet in Fig 3 accordingly:**

```python
import eatpy

**Make the model diagnostic for total chlorophyll available by adding it to the model state.**
**This enables us to assimilate chlorophyll observations.**
experiment = eatpy.models.GOTM(diagnostics_in_state=["total_chlorophyll_calculator_result"])

**Set up ensemble data assimilation using the Error Subspace Transform Kalman Filter**
**(Nerger et al., 2012; https://doi.org/10.1175/MWR-D-11-00102.1)**
filter = eatpy.PDAF(eatpy.pdaf.FilterType.ESTKF)

**Identify biogeochemical state variables by checking for an underscore in their name.**
**(FABM variable names contain at least one underscore; GOTM physical variable names do not)**
bgc_variables = [v for v in experiment.variables if "_" in v]

**Restrict the filter to operating on temperature, salinity and all biogeochemical state variables.**
**Notably, other physical variables such as water velocities and turbulent quantities are**
**thus not affected by assimilation.**
experiment.add_plugin(eatpy.plugins.select.Select(include=["temp", "salt"] + bgc_variables))

**Log-transform all biogeochemical variables. Any associated observations have already been**
**log-transformed in preprocessing (therefore, transform_obs=False)**
experiment.add_plugin(eatpy.plugins.transform.Log(*bgc_variables, transform_obs=False, minimum=1e-12))

**Link remotely sensed surface temperature and chlorophyll observations to their model**
**equivalents. In both cases, the value in the top layer of the model is used.**
**This is the *last* model layer in GOTM, specified by index -1.**
experiment.add_observations("temp[-1]", "cci_sst.dat")
experiment.add_observations("total_chlorophyll_calculator_result[-1]", "cci_chl.dat")

**Run the experiment**
experiment.run(filter)
```

Fig 2 and 3: Mention in the caption that this is Python code.

**We will change the captions accordingly.**

L 258: Are these text files in CSV (comma-separated values) format? Why not use YAML here as well, or a well-structured custom format to avoid/better catch user error?

**Observations must indeed be provided in non-YAML format, specifically, as tab-separated values (TSV). The reason for this is that such files are more suitable for opening/reading as *stream*: EAT does not load the entire observation file in memory but read one line at a time as the simulation progresses. This speeds up initialization, reduces memory consumption, and potentially allows other processes to append to the observation file while the DA experiment is running. TSV is more convenient than YAML for stream reading, as Python's most-used YAML modules (PyYAML, ruamel.yaml) do not expose stream reading as part of their public API. While the benefits of stream reading are modest (in 1D, observation files are usually small, so file read time and memory consumption are minor issues; live addition of observations is uncommon), we feel they are sufficient to prefer TSV over YAML. Moreover, GOTM also uses TSV and TSV-like formats for forcing (e.g. meteorological time series), which means this format would always feature in any EAT experiment. We will summarize this line of reasoning by adding the following:**

**"[with each line describing observation time, observed depth (only for depth-explicit observations), observed value, and its standard deviation] in tab-separated value (TSV) format. This format was chosen over structured formats such as YAML because it enables EAT to read in new observations on-demand while the simulation progresses, instead of having to parse each observation file in its entirety upon start-up."**

Fig 5 and following: "(a)" labels are missing from the figures.

**We will add the missing labels to the figures.**

Fig 5: Science question, only related to the DA result: Are the thin layers of subsurface cooling an effect of including only a few mixing-related sources of uncertainty in the ensemble creation?

**The subsurface cooling appears to be the result of a DA induced change in thermocline depth. As described, the predominant result of data assimilation is a reduction in sea surface temperature and deeper mixing, in particular in early summer. However, there are periods in early autumn where this pattern flips: the DA runs then have a higher surface temperature. This is a critical period during which the surface mixing layer gradually extends deeper into the water column due to surface cooling. In the DA run, the higher surface temperature delays this deepening of the mixing layer. As a result, there is a thin water mass that already lies within the surface mixing layer in the free run, but still below that mixing layer when DA is active. As water masses below the mixing layer [thermocline] are relatively cold, this gives rise to the subsurface cooling signal in autumn.**

**We will summarize this as:**

**"This pattern generally persists into autumn, although occasionally, warmer surface temperatures in the DA experiment cause a decrease in mixing, and accordingly a shoaling in thermocline depth that manifests as subsurface cooling."**

L 322: Is this the result of one "4D" data assimilation cycle with "asynchronous" assimilation, or were multiple cycles performed? Please add this information to the manuscript.

**We will explicitly describe this with the following:**

**"This application uses ensemble-based sequential data assimilation with the Error Subspace Transform Kalman filter (Nerger et al., 2012), using an ensemble of 20 members."**

**Additionally, we will describe assimilation methods more explicitly in table 1 by using "sequential ensemble-based (ESTKF)" and "variational (3D-Var)"**

L 342: Does the assimilation "see" the subsurface chlorophyll maximum, or does the observation operator just work on the top layer of the model?

**It operates on the top layer only. We will add the following when describing the observations that are assimilated:**

**"Both types of observations were mapped to model equivalents in the very top layer of the modelled water column, which is 10 cm thick."**

L 352: "their different components" I would not describe carbon as a "component" of phytoplankton, and would suggest changing it to "their elemental composition" or cell quotas if these are considered.

**We will rephrase this as:**

The BFM model describes the marine lower trophic web through the spatial and temporal evolution of 51 state variables. **BFM uses variable stoichiometry and explicitly represents cycles of carbon, nitrogen, phosphorus, and silicon. Accordingly, it explicitly tracks the fluxes of these elements between its nutrient pools (nitrate, phosphate and silicate) and living functional types (phytoplankton, zooplankton and bacteria)** (Vichi et al., 2020).

L 414: "we further use the "diat-MSP" abbreviation": Initially, I wasn't quite sure what was meant, I'd suggest being more explicit, for example by using "in the following we refer to this parameter as "diat-MSP"".

**We will change the text as proposed.**

**Additional references**

**Andersen, T. K., Bolding, K., Nielsen, A., Bruggeman, J., Jeppesen, E., & Trolle, D. (2021). How morphology shapes the parameter sensitivity of lake ecosystem models. *Environmental Modelling and Software*, *136*(December 2020), 104945. https://doi.org/10.1016/j.envsoft.2020.104945**

**Bannister, R. N. (2017). A review of operational methods of variational and ensemble-variational data assimilation. *Quarterly Journal of the Royal Meteorological Society*, *143*(703), 607–633. https://doi.org/10.1002/qj.2982**

**Gordon, H. R., & McCluney, W. R. (1975). Estimation of the Depth of Sunlight Penetration in the Sea for Remote Sensing. *Applied Optics*, *14*(2), 413–416. https://doi.org/10.1364/ao.14.000413**

**Skákala J, Wakamatsu T, Bertino L, Teruzzi A, Lazzari P, Alvarez E, Cossarini G, Spada S, Nerger L, Vliegen S, Brankart JM, & Brasseur P. (2023). *SEAMLESS Target indicator quality in CMEMS MFCs (D6.1)*. https://doi.org/10.5281/zenodo.10522305**

**Sournia, A. (1982). Is there a shade flora in the marine plankton? *Journal of Plankton Research*, *4*(2), 391–399. https://doi.org/10.1093/plankt/4.2.391**

---

## Author Comment (AC3)

**EAT v0.9.6: a 1D testbed for physical-biogeochemical data assimilation in natural waters**

Jorn Bruggeman, Karsten Bolding, Lars Nerger, Anna Teruzzi, Simone Spada, Jozef Skákala, and Stefano Ciavatta

RC2: 'Comment on gmd-2023-238', Anonymous Referee #2, 31 Jan 2024

Citation: https://doi.org/10.5194/gmd-2023-238-RC2

The authors present a generalized DA framework for 1D ocean applications. The proposed system, EAT, uses GOTM as the physical model, FABM as the biogeochemistry platform and PDAF as the DA software. The authors examined the new system in 3 different locations assimilation various physical and biogeochemical data. They tested state estimation in addition to state and parameter estimation. I believe the system is highly beneficial for the community and quite attractive given its portability and flexibility. The paper is well-written and easy to read. I only have minor comments.

**We would like to thank the referee for the careful review and positive comments on our work. Our point-by-point replies are provided hereafter in green.**

- When generating the initial ensemble, how did the authors decide on the variable distribution (e.g., lognormal) and it's associated parameters? And for the perturbed parameters, I am wondering why did the authors choose k_min and scale_factor over other ones.

**We will include this information by adding the following (new text in bold):**

**"These parameters were selected to showcase the ability to introduce uncertainty in meteorological forcing (u10, v10) as well as the parametrization of the physical and biogeochemical models. Incorporation of other sources of uncertainty in the ensemble will be discussed at the end of this section.** All ensemble members start from the same initial conditions; spread in the ensemble state first seen by the DA filter is generated by simulating 12 hours to the time of the first SST observation.

**For simplicity, the same probability distribution was used to scale all five parameters:** scale factors were drawn from a log-normal distribution with a standard deviation of 0.2 in natural log units. This was done independently per ensemble member, for each of the five scale factors (i.e., they are independently distributed). **The assumption of log-normality is common in biogeochemistry (e.g., Campbell et al., 1995) and ensures that the affected variables remain positive definite; we note, however, that both the type of distribution and its parameters are easy to customize (Fig. 2)."**

**As mentioned in the above, the application will be extended with a paragraph discussing the incorporation of additional uncertainty in the ensemble:**

**"This application can be extended in several ways. For example, additional sources of uncertainty can be introduced when constructing the ensemble. The current setup includes a primitive parametrization of meteorological uncertainty through scaling of the surface wind components; more realistic experiments might source an ensemble of different meteorological model realizations (i.e., separate meteorological forcing files) and distribute those over the EAT ensemble members. EAT facilitates this by allowing its ensemble generators to set YAML parameters (e.g., the location of meteorological forcing in gotm.yaml) to member-specific file paths, similar to how the biogeochemical configuration**

(`fabm/yaml_file`) is treated in Fig. 2. Another option is to introduce uncertainty in biogeochemical parameters other than phytoplankton maximum growth rate. This is easy to realize: all biogeochemical parameters are set in fabm.yaml and any of these can be varied across the ensemble by adding a single line in the ensemble generation script as in Fig. 2. Finally, it is possible to vary physical and biogeochemical initial conditions across the ensemble, as shown in the last section of Fig. 2."

**The updated Fig. 2 is included in our reply to referee comment 1 (RC1).**

- Transforming the state and the data during the update is interesting. I do believe that having Gaussian distributed variables is better suited for Kalman-type correction. My only question is when you transform the actual data, how do you deal with the associated observation error variance. For instance, if I take the logarithm of the data what would be the corresponding error in transformed space? I think the manuscript would benefit from such information.

**We appreciate the opportunity to elaborate on this and will add the following:**

**"All biogeochemical variables were log-transformed to guarantee positivity, as common in biogeochemical data assimilation (e.g., Santana-Falcón et al., 2020; Skákala et al., 2022; Pradhan et al., 2020). Transformation was done with a standard plugin provided by EAT (the Log plugin in Fig. 3), which applies log-transformation to all model variables and by default also to any associated observations. For the latter, the mean and variance of each log-transformed observation is reconstructed from the untransformed mean and variance under the assumption that each observation is perfectly log-normally distributed."**

**For further reference: the transformation logic described in the final sentence is implemented by https://github.com/BoldingBruggeman/eat/blob/797e3f3cad76bbeb2d929fe0b67601b344e164ff/eat py/plugins/transform.py#L42-L53**

- In section 3.1, why not add another experiment assimilating only bgc data? If physical data deteriorates chl then assimilating only chl and adding that to Fig. 6 should be very informative.

**We agree that such an experiment could be informative and propose to add it to the application's Jupyter Notebook and to mention the experiment as possible extension at the end of the application section:**

**"Finally, EAT lends itself well for further experiments that investigate the impact of different types of observations. For instance, an experiment assimilating only surface chlorophyll (included in the application archive; see Data Availability section) could help ascertain whether coupled physical-biogeochemical assimilation performs better or worse than biogeochemistry-only assimilation."**

**We hope the referee will agree this is sufficient, as detailed treatment of the additional experiment and its results would excessively lengthen the application section (already the longest of the three) and complicate its figures. For example, including this additional time series in [already quite busy] Fig 6a makes it difficult to distinguish the time series of the individual experiments:**

[Figure]

- It's very typical in similar 1D applications from the literature to assimilate nutrient profiles. I'm surprised the authors didn't consider that. Is it because the authors don't have access to such data in the tested locations? How about the reanalysis dataset? Addressing subsurface biogeochemical uncertainties can be crucial for adjusting PP across the entire water column (at least within euphotic zone).

**EAT fully supports the assimilation of profiles, but to keep our examples both simple and representative of what can be easily achieved at any location with real observations (time series of remotely sensed temperature and chlorophyll are readily available everywhere), this is not done in the three included applications. We will now discuss the ability to add profiles in detail, as a possible extension of the first application:**

**"Another possible extension is to assimilate observations that describe not just the water surface, but also deeper layers. Notably, the inclusion of depth-explicit biogeochemical observations, e.g., from ship-based casts, automatic profilers or Argo floats, might help determine whether the decrease in subsurface chlorophyll in Fig. 6 is realistic or an artifact of surface-only chlorophyll assimilation. Inclusion of depth-explicit observations in EAT is straightforward, as it merely requires adding a column with depth information to the observation file and dropping the depth index (-1) in the linked model variable (Fig. 3)."**

**In the discussion, we will further note that the included applications demonstrate a subset of EAT's capabilities only:**

**"In general, EAT is flexible: user plugins are given full control over observations, forecasts, and analyses (with the ability to override proposed state updates). Ensemble members can differ in both state and configuration, and ensemble states can be saved and reused through support for restart files. These features can be combined in any number of ways to design new data assimilation experiments. Thus, the applications described here are representative of EAT functionality, but not exhaustive."**

Related to my previous point, I believe assimilating data in the vertical may produce different parameter configurations at different levels beneath the surface.

**This is correct in principle, but vertical variation in biogeochemical parameters is not supported by EAT, as it cannot be implemented without changing the underlying biogeochemical model codes (which explicitly treat parameters as space-invariant scalars). We will note this at the end of the parameter estimation example:**

**"Finally, it is worth noting that in EAT, parameters that are estimated during assimilation remain the same over the entire water column, even though they become variable in time. This is also common in 3D data assimilation (Doron et al., 2013). However, if parameter variability is assumed to stem from shifts in biological species composition, it is worth noting that the current approach cannot account for e.g. separate "light" and "shade" communities (Sournia, 1982), which would require parameter values to vary in depth. This is beyond the scope of EAT; the relevant parameters would need to be added to the depth-explicit model state within the biogeochemical model code."**

Other comments:

- Line 20: that *are* sufficiently

**We will correct this.**

- Line 69: Water column models are *ideal testbeds*

**We will correct this.**

- Line 76: performing *state-parameter* estimation

**We will correct this.**

- Line 172: This allows *the* user

**This section will be rewritten in response to RC1.**

- Line 179: and that *runs* the data

**The subject of this sentence is "scripts", therefore "run" is correct.**

- Line 264: the model *serially*, without

**We will rephrase this as:**

**"Finally, it is also possible to run the model stand-alone, without MPI; in this case, it behaves exactly as the original GOTM-FABM model would."**

- Line 503: different *configuration* options

**We will correct this.**

**Additional references**

**Campbell, J. W. (1995). The lognormal distribution as a model for bio-optical variability in the sea. *Journal of Geophysical Research: Oceans*, *100*(C7), 13237–13254. https://doi.org/10.1029/95JC00458**

Pradhan, H. K., Völker, C., Losa, S. N., Bracher, A., & Nerger, L. (2020). Global Assimilation of Ocean-Color Data of Phytoplankton Functional Types: Impact of Different Data Sets. *Journal of Geophysical Research: Oceans*, *125*(2). https://doi.org/10.1029/2019JC015586

Santana-Falcón, Y., Brasseur, P., Brankart, J. M., & Garnier, F. (2020). Assimilation of chlorophyll data into a stochastic ensemble simulation for the North Atlantic Ocean. *Ocean Science*, *16*(5), 1297–1315. https://doi.org/10.5194/os-16-1297-2020

Skákala, J., Bruggeman, J., Ford, D., Wakelin, S., Akpınar, A., Hull, T., Kaiser, J., Loveday, B. R., O'Dea, E., Williams, C. A. J., & Ciavatta, S. (2022). The impact of ocean biogeochemistry on physics and its consequences for modelling shelf seas. *Ocean Modelling*, *172*, 101976. https://doi.org/10.1016/j.ocemod.2022.101976

Sournia, A. (1982). Is There a Shade Flora in the Marine Plankton. *Journal of Plankton Research*, *4*(2), 391–399. https://doi.org/10.1093/plankt/4.2.391

---

## Author Response (AR2)

**Authors' response to anonymous reviewer #1**

**Jorn Bruggeman on behalf of all authors**

The authors addressed my previous comments, and I like the suggested extensions which have been added to the description of the three EAT example applications. The manuscript is easy to follow, and the modified code snippets (Fig. 2, 3, 5) are commented well and easier to interpret. I have only a few minor comments and a bug report related to the first example application.

**We much appreciate the reviewer taking the time to reread the ms, rerun the example applications and provide comments. As described below, we have adopted nearly all suggestions verbatim.**

**bug report**

This time around, I tried out all the example applications. Overall, I like the use of Jupyter notebooks, which make it straightforward to run the examples and visualize the output. Perhaps one could reduce the GOTM text output a bit, which takes a lot of scrolling to get through.

Unfortunately, I ran into an early error in the first application "Ensemble" in cell 3 which then lead to a python error in the following cell (No such file or directory: 'result.nc').

Hopefully, this output will help find the bug that may be remaining (I updated eatpy before running the example and hope the problem is not on my end):

```
model(no filter program present)

initialize_gotm
* * *
Reading configuration from: gotm.yaml

configuring modules ....

init_airsea_yaml

done

[...]

Initializing ice...

model type: surface_constant

initialization succeeded.

ERROR: the following setting(s) were not recognized:

- /instances/phy/coupling/fer

FATAL ERROR: initialize: invalid configuration

STOP 1
```

```

The two other applications ran fine on my machine.

**We suspect that this error was due to EAT being out of date on the reviewer's system, as the error message indicates that a new biogeochemical configuration (fabm.yaml) is being used with a previous version of the PISCES model. The updated PISCES model was included in the 0.9.8 EAT release (https://doi.org/10.5281/zenodo.10934071) that accompanied our revision; moreover, the combination of this 0.9.8 release and the updated applications was extensively tested before we submitted the revised ms. The reviewer does indicate that he/she tried to update EAT before retesting, but we speculate that this may have failed due to a change in the location of EAT packages (they moved from our private bolding-bruggeman channel to the public conda-forge channel).**

**In any case, we have updated both EAT and the example applications again for this new revision, mainly to improve the file and folder organization of the applications. The new release has again been extensively tested (https://github.com/BoldingBruggeman/eat-paper-applications/actions/runs/8939243820/job/24560968244), and we now also include a note in the application description stating which version of EAT is required (https://doi.org/10.5281/zenodo.10307315). This should ensure that such problems do not recur.**

**The update of EAT is reflected by the version increase in the ms title (now: v1.0.0).**

**specific comments (line numbers are based on the "tracked changes" version of the manuscript)**

L 57 + 58: In DA lingo, 3D DA is often distinguished from 4D DA, where both use 3D models. Because this part of the manuscript is referring to DA applied to 3D models, no matter if 3D or 4D DA is used, I would recommend using "data assimilation systems based on 3D models", rather than the current "3D data assimilation systems", just to avoid confusion.

**Done**

L 199: Here a double dash "--" accidentally turned into an em dash "–".

**Corrected**

Fig 5: This looks nice and short for implementing something rather complex! Maybe mention that "before_analysis" is run just before each assimilation update (I presume).

**Added**

L 329: Why not mention the strong coupling in the second experiment as a contrast to the first experiment where the weak coupling is emphasized in the text?

**We have added "(strong coupling: assimilation is applied simultaneously to the full system state, with covariances between physical and biogeochemical components allowed to be non-zero)"**

L 335: "which applies log-transformation to all model variables": I presume only the biogeochemical variables are log-transformed, please make this explicit by either adding "biogeochemical", "physical and biogeochemical" or similar.

**That is correct. As per the preceding sentence, "all *biogeochemical* variables were log-transformed". The subsequent sentence describes generic functionality of the plugin, which could in principle be applied to any selection of physical and/or biogeochemical variables. We now clarify this by stating "which applies log-transformation to a user-specified subset of model variables"**

L 346: An early reference to Fig 2 might be useful here.

**We have added "This was implemented using EAT's built-in support for ensemble generation (Fig 2)."**

Fig 7: It might be confusing for new readers to have the same experiment described in 3 different ways in the same figure: "phys+bgc DA", "remotely sensed temperature and chlorophyll were assimilated", "both SST and chlorophyll were assimilated". I would suggest a more consistent description, which could also reduce the length of the caption.

**Changed as suggested**

L 569: "This is beyond the scope of EAT; the relevant parameters would need to be added to the depth-explicit model state within the biogeochemical model code.": While true, I would suggest reframing it slightly and simply mentioning that such a change would require modification to the code of the biogeochemical model.

**We now write: "This can only be achieved by modifying the (Fortran) source code of the biogeochemical model in order to replace each affected (scalar) parameter by a spatially resolved state variable."**